 **eLIFE**

# The adhesion-GPCR BAI1 shapes dendritic arbors via Bcr-mediated RhoA activation causing late growth arrest

Joseph G Duman[1]*, Shalaka Mulherkar[1], Yen-Kuei Tu[2], Kelly C Erikson[1], Christopher P Tzeng[1†], Vasilis C Mavratsas[1,3‡], Tammy Szu-Yu Ho[4§], Kimberley F Tolias[1,2,5]*

[1]Department of Neuroscience, Baylor College of Medicine, Houston, United States; [2]Graduate Program in Integrative Molecular and Biomedical Sciences, Baylor College of Medicine, Houston, United States; [3]Rice University, Houston, United States; [4]Program in Developmental Biology, Baylor College of Medicine, Houston, United States; [5]Verna and Marrs McLean Department of Biochemistry and Molecular Biology, Baylor College of Medicine, Houston, United States

**\*For correspondence:**
duman@bcm.edu (JGD);
tolias@bcm.edu (KFT)

**Present address:** [†]Department of Biology, Stanford University, Stanford, United States; [‡]School of Medicine, University of Texas Medical Branch, Galveston, United States; [§]Departments of Neurology and Neurobiology, Harvard Medical School, Boston Children's Hospital, Boston, United States

**Competing interests:** The authors declare that no competing interests exist.

**Abstract** Dendritic arbor architecture profoundly impacts neuronal connectivity and function, and aberrant dendritic morphology characterizes neuropsychiatric disorders. Here, we identify the adhesion-GPCR BAI1 as an important regulator of dendritic arborization. BAI1 loss from mouse or rat hippocampal neurons causes dendritic hypertrophy, whereas BAI1 overexpression precipitates dendrite retraction. These defects specifically manifest as dendrites transition from growth to stability. BAI1-mediated growth arrest is independent of its Rac1-dependent synaptogenic function. Instead, BAI1 couples to the small GTPase RhoA, driving late RhoA activation in dendrites coincident with growth arrest. BAI1 loss lowers RhoA activation and uncouples it from dendrite dynamics, causing overgrowth. None of BAI1's known downstream effectors mediates BAI1-dependent growth arrest. Rather, BAI1 associates with the Rho-GTPase regulatory protein Bcr late in development and stimulates its cryptic RhoA-GEF activity, which functions together with its Rac1-GAP activity to terminate arborization. Our results reveal a late-acting signaling pathway mediating a key transition in dendrite development.
DOI: https://doi.org/10.7554/eLife.47566.001

## Introduction

A dizzying array of dendritic arbors receives and processes information transiting the brain (*Harris and Spacek, 2016*; *Landgraf and Evers, 2005*; *Lefebvre et al., 2015*; *Polavaram et al., 2014*). Arbor morphology controls neuronal availability to inputs (*Landgraf and Evers, 2005*; *Lefebvre et al., 2015*; *Häusser and Mel, 2003*) and contributes to input processing (*Häusser and Mel, 2003*; *Grienberger et al., 2015*; *Henze et al., 1996*; *London and Häusser, 2005*). Thus, arbor malformation or degradation manifests in numerous neurodevelopmental and neurodegenerative diseases (*Kulkarni and Firestein, 2012*). Due to the plethora of dendritic arbor morphologies, many of which are highly elaborate, arbor construction represents a significant cell biological challenge. Intrinsic genetic programs are important, but insufficient, for instructing neurons to form correct dendrite architecture, and extrinsic signals and signaling pathways must supply critical environmental information (*Lefebvre et al., 2015*; *Dong et al., 2015*).

Final arbor structure arises from a tension between drivers and restrictors of dendritic growth (*Lefebvre et al., 2015*; *Dong et al., 2015*). Restrictor failure leads to dendritic overgrowth, while ectopic restrictor activation causes dendrite retraction. Many studies have shown alterations in

 

brain size (both macro- and microcephaly) in patients with autism spectrum disorder (ASD), particularly affecting the hippocampus, which functions in spatial learning, memory consolidation, and emotional control (*Kulkarni and Firestein, 2012*; *Copf, 2016*; *Fombonne et al., 1999*; *Lainhart et al., 1997*). Dendrites account for at least 30–40% of gray matter volume (*Curran et al., 2017*; *Karbowski, 2015*; *Kasthuri et al., 2015*), so defects in arbor size could contribute to such brain defects. Indeed, dendritic hypertrophy is observed in the hippocampi of mouse models of ASD (*Cheng et al., 2017*; *Kwon et al., 2006*), and dendritic hypotrophy characterizes other forms of ASD (*Kulkarni and Firestein, 2012*; *Kweon et al., 2017*). Further, humans and mice suffering from chronic stress, depression/depressive-like behavior, or neurodegenerative disease exhibit dendritic hypotrophy (*Kulkarni and Firestein, 2012*; *de Flores et al., 2015*; *Zhao et al., 2017*).

ASD-associated macrocephaly manifests well after birth (*Lainhart et al., 1997*; *Stigler et al., 2011*), which implicates the dysfunction of late-acting dendrite restriction mechanisms. While recent work has begun to illuminate the molecular nature of the stable state of dendrites subsequent to growth (*Kerrisk et al., 2013*; *Lin et al., 2013*), little is known about the events whereby neurons in the hippocampus make this transition. The small GTPase Rem2 (*Ghiretti et al., 2013*), the leucine-rich repeat protein Lrig1 (*Alsina et al., 2016*), and the adhesion-GPCR (A-GPCR) CELSR3 (*Shima et al., 2007*) all contribute to apparently separate pathways that act as a brake on hippocampal dendrite growth in response to various stimuli during early postnatal development, but almost nothing is known about how these neurons transition out of a net growth state to the more mature, stable state.

Brain-specific angiogenesis inhibitor 1 (BAI1, also ADGRB1(28)) is an A-GPCR that instigates phagocytosis of apoptotic cells and Gram-negative bacteria (*Das et al., 2011*; *Elliott et al., 2010*; *Park et al., 2007*). BAI1 is expressed in the cortex, hippocampus, striatum, and thalamus in neurons and glia (*Sokolowski et al., 2011*) and suppresses brain angiogenesis (*Kaur et al., 2005*), its loss promoting glioblastoma (*Koh et al., 2004*; *Zhu et al., 2011*). BAI1 also regulates dendritic spine formation, excitatory synaptogenesis and synaptic plasticity in the hippocampus and cortex (*Duman et al., 2013*; *Zhu et al., 2015*; *Tu et al., 2018*). We demonstrated that BAI1 promotes spino- and synaptogenesis by recruiting the Par3/Tiam1 polarity complex to spines, resulting in localized activation of the small GTPase Rac1 and actin cytoskeletal remodeling (*Duman et al., 2013*). Moreover, we recently reported that BAI1 functionally interacts with the ASD-associated synaptic organizer Neuroligin-1, and coordinates trans-synaptic signaling important for excitatory synaptogenesis (*Tu et al., 2018*). Notably, the BAI1 gene is located in a hot spot for de novo germline mutations in human ASD patients (*Michaelson et al., 2012*), and BAI1 SNPs associate with educational attainment and mathematical ability (*Lee et al., 2018*). Moreover, BAI1 mRNA is elevated in mouse models of Rett and MeCP2 duplication syndromes, both of which exhibit altered dendritic arbors and ASD-like symptoms (*Chahrour et al., 2008*; *Urdinguio et al., 2008*).

We report here that BAI1 also functions to restrict the growth of dendritic arbors in hippocampal pyramidal cells. This restriction happens late in development as dendritic arbors transition from growth to stabilization. While no known modes of BAI1 signaling account for its ability to restrict dendritic growth, we show that it activates the small GTPase RhoA by stimulating the cryptic RhoA-guanine nucleotide exchange factor (GEF) activity of the breakpoint cluster region protein (Bcr), and that Bcr's activation of RhoA is required in concert with its inhibition of the small GTPase Rac1 to mediate growth arrest. This previously unidentified pathway represents a potential clinical target for diseases that alter dendritic arbor form.

## Results

### BAI1 loss results in late developmental dendritic overgrowth

We have shown that BAI1 promotes spinogenesis and excitatory synaptogenesis in cultured hippocampal neurons and in vivo in mouse cortex and hippocampus (*Duman et al., 2013*; *Tu et al., 2018*). To determine the contribution of BAI1 to dendritic arbor development, we introduced an shRNA against *Adgrb1* (*Duman et al., 2013*; *Tu et al., 2018*; *Figure 1—figure supplement 1*) and enhanced green fluorescent protein (EGFP) into embryonic day 14 (E14) mice via in utero electroporation and assayed the effects at postnatal day 21 (P21). The shRNA lowered BAI1 expression in neuronal somata to undetectable levels, but spared non-transfected cells (*Figure 1a*). Use of shRNA to

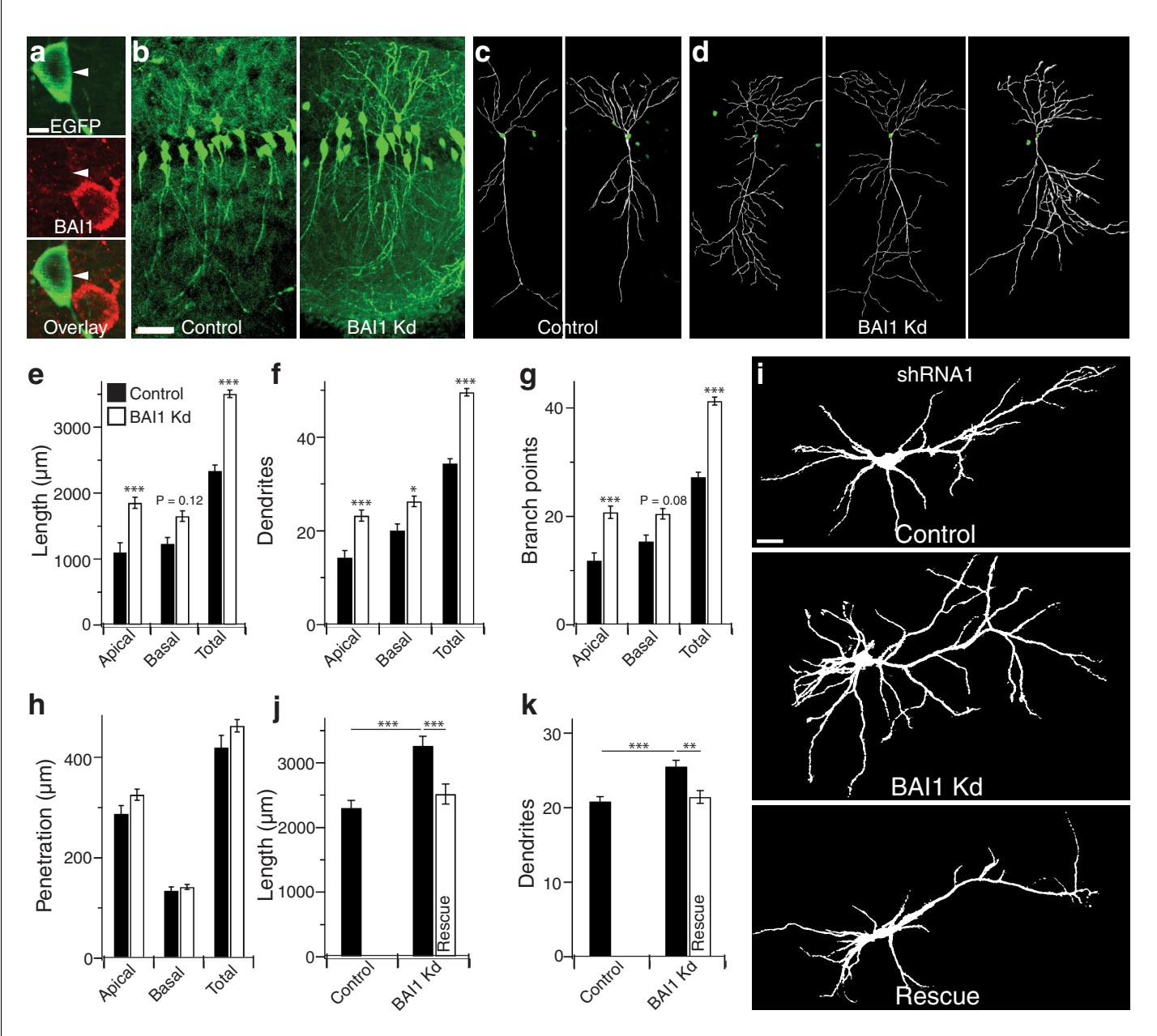

**Figure 1.** BAI1 restricts dendrite growth in the developing hippocampus and hippocampal cultures. (a) Representative in vivo images showing optical sections through the centers of neuronal somata from mice transfected with EGFP and shRNA against *Adgrb1* and stained for BAI1. Note that the transfected neuron expressing EGFP and shRNA has substantially lower BAI1 staining that the neighboring untransfected neuron, indicating successful BAI1 knockdown. (b) 100 μm thick slices of dorsolateral CA1 from mice transfected with EGFP (Control) or EGFP and shRNA against BAI1 (BAI1 Kd). Bar is 50 μm. c and d Reconstructions of representative control (c) and BAI1 Kd (d) dendritic arbors from slices like those in b. (e) Summary data for dendrite length. (f) Summary data for dendrite number. (g) Summary data for branch points. (h) Summary data for dendrite penetration. Total number of neurons: 21 (N = 3) for control and 41 (N = 4) for BAI1 Kd. (i) Representative images of neurons expressing EGFP alone (control) or with shRNA against *Adgrb1* (BAI1 Kd) or with shRNA against BAI1 and RNAi-resistant BAI1 (Rescue). Images are masked to remove axons and dendrites from other neurons. Bar is 20 μm. (j) Summary data for total length of cultured neurons. (k) Summary data for total dendrite number in cultured neurons. Total number of neurons in j and k: 118 for control neurons, 106 for BAI1 Kd neurons, and 65 for rescue neurons (N = 7). Data are represented as mean ± s. e.m. (***p<0.001) Detailed statistics are found in *Figure 1—source data 1*.

DOI: https://doi.org/10.7554/eLife.47566.002

The following source data and figure supplements are available for figure 1:

**Source data 1.** Statistical summary for *Figure 1*: ANOVA and key Tukey *post-hoc* tests and N and n values for *Figure 1e–h,j–k*.

*Figure 1 continued on next page*

*Figure 1 continued*

DOI: https://doi.org/10.7554/eLife.47566.010

**Source data 2.** Individual values for in vivo and cultured neurons demonstrating the effects of BAI1 loss on arbor growth (*Figure 1e–h,j–k*).

DOI: https://doi.org/10.7554/eLife.47566.007

**Source data 3.** Quantification of Western bands showing expression levels of BAI1, BAI2, and BAI3 (*Figure 1—figure supplement 1d*).

DOI: https://doi.org/10.7554/eLife.47566.008

**Source data 4.** Extended individual values for in vivo and cultured neurons showing the effects of BAI1 loss on arbor growth (*Figure 1—figure supplement 1a,b,e,g,h,i,j*).

DOI: https://doi.org/10.7554/eLife.47566.009

**Figure supplement 1.** shRNA-mediated BAI1 Kd and effects on other BAIs.

DOI: https://doi.org/10.7554/eLife.47566.003

**Figure supplement 1—source data 1.** Statistical summary for *Figure 1—figure supplement 1*: ANOVA and key Tukey *post-hoc* tests and N and n values for *Figure 1—figure supplement 1d*.

DOI: https://doi.org/10.7554/eLife.47566.004

**Figure supplement 2.** BAI1 restricts dendrite growth in developing hippocampus.

DOI: https://doi.org/10.7554/eLife.47566.005

**Figure supplement 2—source data 1.** Statistical summary for *Figure 1—figure supplement 2*: ANOVA and key Tukey *post-hoc* tests and N and n values for *Figure 1—figure supplement 2a,b,e,g–j*.

DOI: https://doi.org/10.7554/eLife.47566.006

lower BAI1 levels is hereafter referred to as BAI1 Kd. The dorsolateral hippocampal CA1 region exhibited a striking increase in dendrite density in BAI1 knockdown (Kd) brains relative to controls, though transfection levels were similar (*Figure 1b*). We reconstructed the dendritic arbors of neurons whose somata resided in the center of 100 μm slices cut from control (*Figure 1c*) and BAI1 Kd (*Figure 1d*) brains, revealing that BAI1 Kd resulted in longer arbors (*Figure 1e*), increased dendrite number (*Figure 1f*), and increased branch points (*Figure 1g*) relative to controls. Though BAI1 Kd arbors overgrew, they showed no excess dorsoventral penetration (*Figure 1h*), indicating that they were still subject to hippocampal boundaries. BAI1 Kd arbors were also wider than those of control neurons (*Figure 1—figure supplement 2a,b*). The effects of BAI1 Kd were apparent in both apical and basal dendrites (*Figure 1e–g*, *Figure 1—figure supplement 2a,b*), though apical dendrites showed greater sensitivity to BAI1 loss (*Figure 1—figure supplement 2c*). We also observed defects in the orientation of primary apical dendrites from BAI1 Kd neurons (*Figure 1—figure supplement 2d,e*). Thus, sparse in vivo loss of BAI1 expression causes striking mis- and overgrowth of hippocampal pyramidal neuron dendritic arbors.

We tested whether dendritic overgrowth also manifests in primary hippocampal cultures. *Figure 1i* shows the somatodendritic domains of representative cultured neurons from control, BAI1 Kd, and BAI1 Kd neurons expressing RNAi-resistant wild-type BAI1 (Rescue). The effects of BAI1 Kd on arbor length (*Figure 1j*), dendrite number (*Figure 1k*), and dendritic arbor complexity (as measured by Sholl analysis, *Figure 1—figure supplement 2i*) revealed that BAI1 Kd also caused dendrite overgrowth in neuronal cultures and that this was rescued by reintroduction of BAI1. Arbor overgrowth was also apparent when a separate, non-overlapping shRNA (*Figure 1—figure supplement 1*) directed against BAI1 was introduced into cultured neurons, and these defects were also rescued by RNAi-resistant wild type BAI1 (*Figure 1—figure supplement 2f–h,j*). Thus, BAI1 Kd with either of two shRNAs in cultured hippocampal neurons reproduced the in vivo effects of BAI1 Kd.

To gain insight into BAI1-mediated dendritic arbor restriction, we determined the time courses of BAI1 Kd and arbor growth. Initially, shRNAs were introduced into hippocampal neurons at 7 days in vitro (DIV), and neurons were fixed and stained at various times. Some loss of BAI1 was apparent in BAI1 Kd neurons by 10 DIV, becoming complete at 14 DIV and persisting though at least 21 DIV (*Figure 2—figure supplement 1a*). Significant arbor growth occurred in all neurons between 10 and 17 DIV; however, control neurons stopped growing at 17 DIV, while BAI1 Kd neurons continued to grow through at least 21 DIV (*Figure 2—figure supplement 1b,c*). We also performed longitudinal imaging of neurons into which shRNA was introduced at 6 DIV and obtained similar results. At 10 DIV, BAI1 Kd neurons had fewer and shorter dendrites than controls, but they surpassed control neurons in dendrite tip number by DIV 14 and in length by 17 DIV (*Figure 2a–c*). The longitudinal nature of these experiments allowed measurement of branch dynamics. Branch formations and

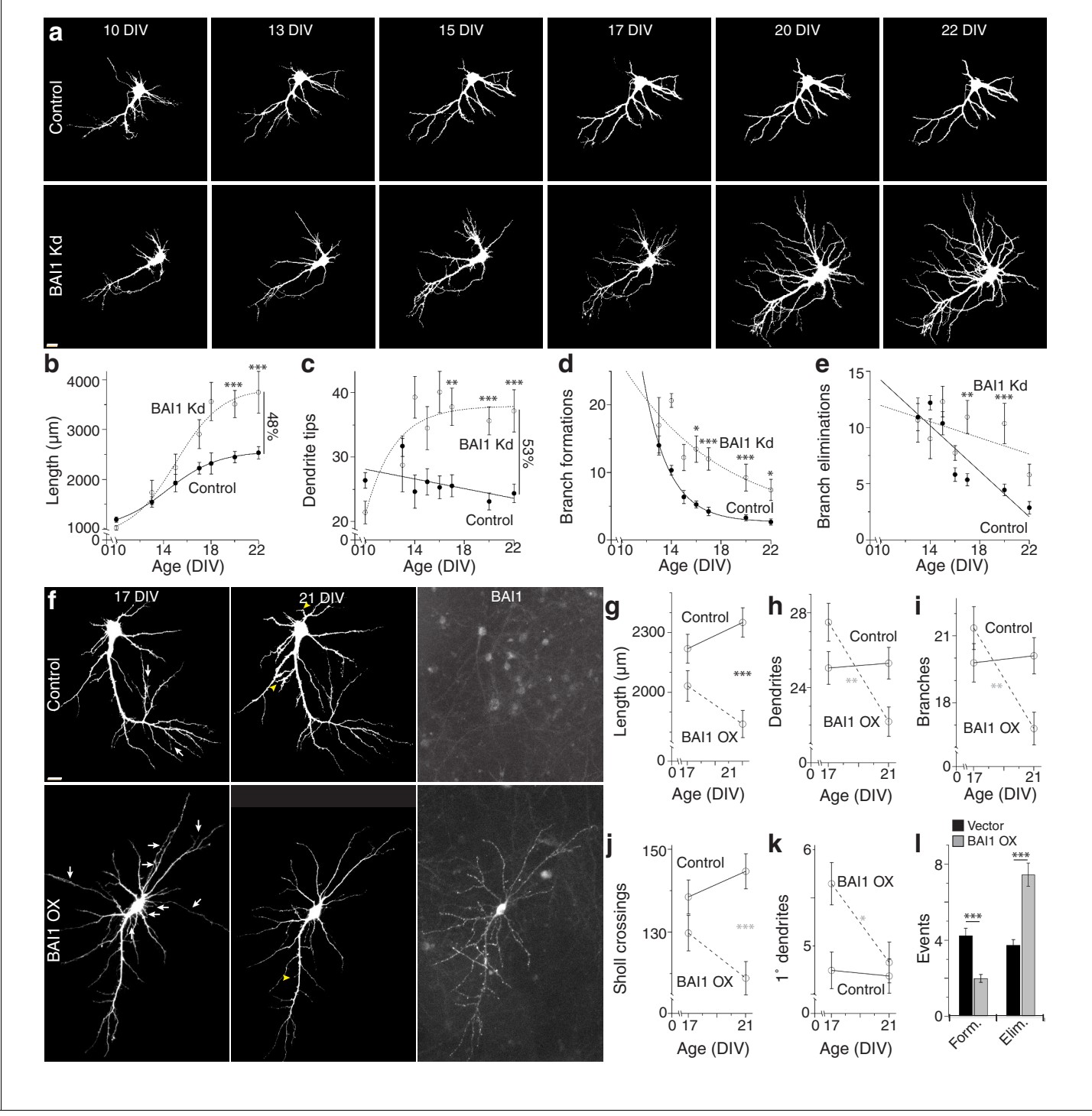

**Figure 2.** BAI1 exercises its effects on dendritic growth arrest late in development. (a) Representative longitudinal series from neurons transfected on 6 DIV with EGFP (Control) or EGFP and shRNA against *Adgrb1* (BAI1 Kd) imaged at the indicated times. Images are masked to remove axons and dendrites from other neurons. Bar is 10 μm. (b) Summary data of total dendrite length. (c) Summary data of total dendrite number. (d) Summary data of branch formations. (e) Summary data of branch eliminations. Total number of neurons represented in panels a–e): 61 for control neurons and 22 for BAI1 Kd neurons (N = 5). Data are represented as mean ± s.e.m. with mathematical fits. (f) Representative longitudinal images of neurons expressing EGFP alone (Control) or with overexpression of *Adgrb1* (BAI1 OX). Neurons were fixed on 21 DIV and immunostained for BAI1. White arrows in the 17 DIV panels indicate dendrites eliminated by 21 DIV, while yellow arrowheads in 21 DIV panels indicate dendrites formed since 17 DIV. Images are masked to remove axons and dendrites from other neurons. Bar is 10 μm. (g) Summary data for dendrite length. (h) Summary data for dendrite number. (i) Summary data for branch number. (j) Summary data for integrated Sholl crossings. (k) Summary data for primary dendrites. (l) Summary data for

*Figure 2 continued on next page*

*Figure 2 continued*

dendrite dynamics. Total number of neurons represented in panels (f–l): 105 neurons (N = 3). Symbols positioned between points indicate differences between populations at the different time points. Data are represented as mean ± s.e.m. (***p<0.0001, **p<0.01, *p<0.05) Detailed statistics are found in *Figure 2—source data 1*.

DOI: https://doi.org/10.7554/eLife.47566.011

The following source data and figure supplements are available for figure 2:

**Source data 1.** Statistical summary for *Figure 2*: ANOVA and key Tukey *post-hoc* tests and N and n values for *Figure 2c–e,g–l*.
DOI: https://doi.org/10.7554/eLife.47566.016

**Source data 2.** Individual values for BAI1 Kd and OX time course experiments (*Figure 2b–e,g–l*).
DOI: https://doi.org/10.7554/eLife.47566.014

**Source data 3.** Individual values for arbor growth in fixed time courses and BAI1 subcellular staining levels (*Figure 2—figure supplement 1c,h*).
DOI: https://doi.org/10.7554/eLife.47566.015

**Figure supplement 1.** Manipulation of BAI1 leads to altered dendrite growth in fixed populations of neurons; BAI1 localizes to dendritic tips later in development.
DOI: https://doi.org/10.7554/eLife.47566.012

**Figure supplement 1—source data 1.** Statistical summary for *Figure 2—figure supplement 1*: ANOVA and key Tukey *post-hoc* tests and N and n values for *Figure 2—figure supplement 1 c, f*.
DOI: https://doi.org/10.7554/eLife.47566.013

eliminations decreased over the 10–22 DIV time course in control and BAI1 Kd neurons, but more slowly in BAI1 Kd neurons (*Figure 2a,d,e*). These data show that BAI1 loss removes a late-acting brake on arbor development, leading to longer, more branched, and less stable dendritic arbors.

We previously demonstrated that BAI1 is enriched in hippocampal dendritic spines (*Duman et al., 2013*). To determine BAI1 expression in other loci germane to dendrite development, we examined BAI1-staining in the neurons used in *Figure 2—figure supplement 1a–c*. We were especially interested in dendrite tips and branch points, which contribute to dendrite growth dynamics (*Dong et al., 2015*). BAI1 was present at branch points at 14, 17, and 21 DIV, but did not vary relative to total dendritic BAI1 staining over this time course (*Figure 2—figure supplement 1d–f*). This differed strikingly from the behavior of BAI1 at dendritic tips. 14 DIV neurons exhibited little BAI1 staining at the dendritic tips, but clear BAI1 staining appeared by 17 DIV and persisted through at least 21 DIV (*Figure 2—figure supplement 1d–f*). Thus, BAI1 is present throughout dendrites and critical growth loci at the time of dendrite growth arrest.

Our results demonstrate that BAI1 loss leads to late dendritic arbor overgrowth. However, is the converse true, i.e. does late BAI1 overexpression (OX) cause arbor retraction? To test this, we overexpressed BAI1 in cultured neurons at 6 DIV and took longitudinal images at 17 and 21 DIV. BAI1 OX was distributed throughout the somatodendritic domains of transfected neurons (*Figure 2f*). BAI1 OX and control neurons exhibited similar length, dendrite and branch number, and integrated Sholl crossings at 17 DIV, and these parameters remained fairly constant at 21 DIV in control neurons (*Figure 2f–j*). In contrast, all of these parameters had decreased by 21 DIV in BAI1 OX neurons (*Figure 2f–j*). Additionally, BAI1 OX neurons exhibited one more primary dendrite than control neurons at 17 DIV that disappeared by 21 DIV (*Figure 2f,k*). In agreement with the data in *Figure 2d–e*, branch formation and elimination was essentially balanced in control neurons from 17 to 21 DIV, whereas BAI1 OX decreased branch formation and increased branch elimination (*Figure 2f,l*). Taken together, *Figures 1–2* suggest that BAI1 mediates the transition from arbor growth to termination and arbor stabilization.

## BAI1 connects RhoA to normal dendritic growth arrest

How does BAI1 cause dendritic arbor growth arrest? One obvious candidate mechanism is via inhibition of the small GTPase Rac1, which promotes dendrite growth and branching (*Dong et al., 2015*; *Park et al., 2007*; *Evers et al., 2000*; *Li et al., 2015*; *Sin et al., 2002*; *Wong et al., 2000*). We investigated this possibility by measuring long-term Rac1 activation (*Figure 3—figure supplement 1*) in control and BAI1 Kd neurons with a Raichu-Rac1 FRET probe (*Komatsu et al., 2011*), but determined that this was probably not the case, as differences in Rac1 activation in BAI1 Kd neurons were not those expected to drive dendrite growth (*Figure 3—figure supplement 2a–c*). Another potential mechanism of BAI1-mediated growth arrest is activation of the small GTPase RhoA, which

inhibits dendrite arbor elaboration (*Redmond and Ghosh, 2001*). We therefore used the Raichu-RhoA FRET probe (*Nakamura et al., 2006*) to measure RhoA activation in control and BAI1 Kd neurons (*Figure 3a*). Strikingly, control neurons exhibited a marked increase in RhoA activation at dendrite tips (*Figure 3a,b*) and branch points (*Figure 3a,c*) late in development. BAI1 Kd neurons, by contrast, showed blunted and delayed RhoA activation at the end of the 15–23 DIV developmental window (*Figure 3a–c*). These data reveal that BAI1 mediates a heretofore-undescribed late RhoA activation accompanying dendritic growth arrest.

That BAI1 signals to RhoA to effect a transition out of dendrite growth is an intriguing hypothesis, but the data linking RhoA to dendrite growth arrest are incomplete. To address this, we assayed dendrite arbor growth in hippocampal neurons from RhoA$^{flox/flox}$ mice (*Mulherkar et al., 2013*; *Mulherkar et al., 2014*). These experiments revealed that RhoA KO neurons exhibited increased arbor size and complexity relative to controls late in development (19 DIV) (*Figure 3—figure supplement 3a–e*). But does RhoA activation drive dendrite behavior, and how does BAI1 affect this relationship? To answer these questions, we analyzed dendrites persistent from 16 to 21 DIV in hippocampal neurons expressing the RhoA FRET probe RhoA-flare (*Pertz et al., 2006*). We classified all persistent dendrites as stable (<10% change in length), extending (≥10% net growth), or retracting (≥10% net shrinkage). The proportions of these pools differed greatly between control and BAI1 Kd neurons: extending dendrites formed twice the share of total dendrites in BAI1 Kd neurons as in control neurons, with the reverse being true for retracting neurons; the proportion of stable neurons was essentially the same (*Figure 3d*). In control neurons, RhoA activation correlated strikingly with individual dendrite behavior (*Figure 3e,g,h*): extending dendrites showed low levels of RhoA activation, while retracting dendrites had high levels thereof. These differences in RhoA activation were persistent throughout the time course and especially pronounced at the dendrite tips (*Figure 3g*, left), though also apparent at branch points (*Figure 3h*, left). BAI1 Kd profoundly affected these phenomena. BAI1 Kd neurons had lower RhoA activation levels throughout the time course (*Figure 3f–h*, right). At dendrite tips, which were the best indicators of dendrite behavior in control neurons, there was no difference in RhoA activation between extending and retracting dendrites in BAI Kd neurons (*Figure 3f,g*). The differences between RhoA activation in extending and retracting dendrites were maintained at branch points in BAI1 Kd neurons, though RhoA activation was generally lower than in control neurons (*Figure 3f–h*, right). These differences in RhoA activation do not merely arise from differences in dendrite behavior because within the stable, extending, and retracting dendrite populations, relative dendrite length changes were the same in control and BAI1 Kd neurons (*Figure 3—figure supplement 2d*).

The population data for control neurons in *Figure 3g&h* suggested that by 16 DIV, most persistent dendrites have established their fate with regard to extension or retraction. We gathered all of the individual dendrite growth/retraction events from our time courses into control and BAI1 Kd pools and sought to determine an empirical, quantitative rule that links RhoA activation to arbor behavior. The best match was shown in *Figure 3i*, wherein RhoA activation levels at dendrite tips strongly and negatively correlated with dendrite growth. BAI1 Kd, however, completely eliminated this correlation (*Figure 3i*). Correlation of dendrite growth with a number of other parameters was less informative (*Figure 3—figure supplement 2e–g*). These data suggest that RhoA activation at dendritic tips is the primary driver of dendritic retraction in wild type neurons.

## BAI1 effects on dendritic arbors are rescued by compensatory changes in RhoA signaling

If BAI1 functions through RhoA to cause dendrite growth arrest, forced RhoA activation should compensate for BAI1 Kd. We tested this using narciclasine, which activates RhoA and is not neurotoxic up to 100 nM (*Krug et al., 2013*). We first confirmed that narciclasine activated RhoA in neurons by treating Raichu-RhoA-expressing hippocampal neurons with vehicle or 10 nM narciclasine. Narciclasine-treated neurons exhibited increased RhoA activation throughout the dendritic arbor (*Figure 4—figure supplement 1a,b*). We therefore applied narciclasine to 17 DIV neurons and found that this low dose of narciclasine had no effect on the morphology of control neurons assayed on 21 DIV, but prevented BAI1 Kd-mediated dendrite overgrowth with respect to dendrite length (*Figure 4a,b*), dendrite number (*Figure 4a,c*), branch points (*Figure 4a,d*), and Sholl curves (*Figure 4—figure supplement 1c,d*). Like BAI1 Kd, narciclasine had no effect on primary dendrite number (*Figure 4a,e*). We also examined spine density and morphology under these conditions. As we have previously

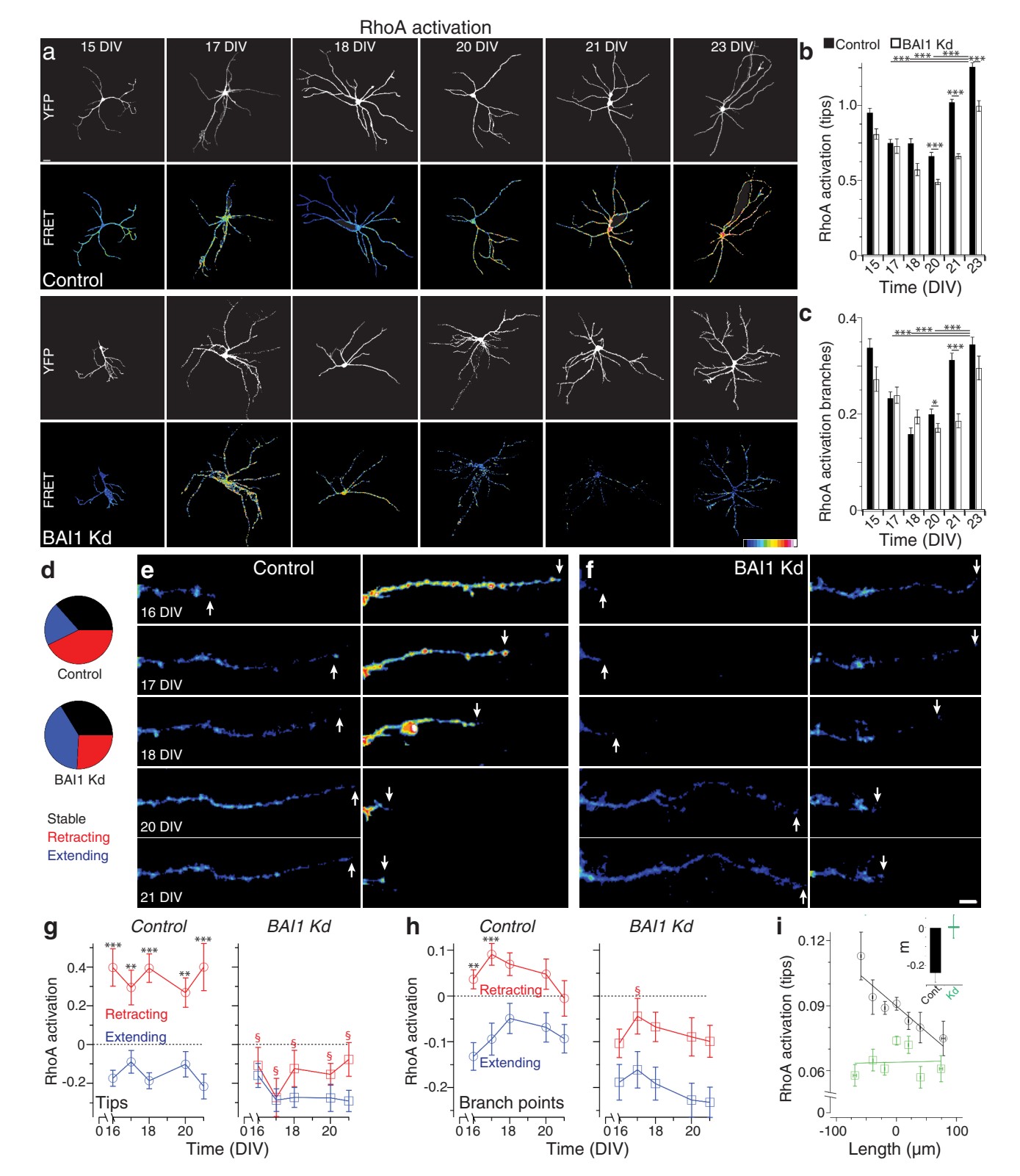

**Figure 3.** BAI1 drives a late dendritic RhoA activation peak and ties RhoA activation to dendrite behavior. (**a**) Representative images of hippocampal neurons expressing Raichu-RhoA alone (control) or with shRNAs against *Adgrb1* (BAI1 Kd) measured at the indicated times. Overall dendrite structure was ascertained by directly exciting the FRET acceptor (YFP images), while relative RhoA activation is shown in the FRET images according to the color code in the lower right corner (low values on the left, high on the right). Images are masked to remove axons and dendrites from other neurons. Bar is

*Figure 3 continued on next page*

*Figure 3 continued*

20 µm. (**b**) Summary data of RhoA activation at dendrite tips. (**c**) Summary data of RhoA activation at dendrite branch points. Total number of neurons: 49 (15 DIV, control), 27 (15 DIV, BAI1 Kd), 34 (17 DIV, control), 20 (17 DIV, BAI1 Kd), 35 (18 DIV, control), 34 (18 DIV, BAI1 Kd), 36 (20 DIV, control), 38 (20 DIV, BAI1 Kd), 37 (21 DIV, control), 32 (21 DIV, BAI1 Kd), 53 (23 DIV, control), and 31 (23 DIV, BAI1 Kd) (N = 5). (**d**) Pie charts showing the behavior of hippocampal dendrites persistent from 16 to 21 DIV in Control and BAI1 Kd neurons. Throughout this figure, stable dendrites are shown in black, retracting dendrites in red, and extending dendrites in blue. e and f Representative longitudinal images of extending (left) and retracting (right) dendrites in Control (**e**) and BAI1 Kd (**f**) neurons expressing the RhoA-FLARE reporter. The color scale for RhoA activation is the same as in panel A. Arrows track dendrite tip location. Images are masked to remove axons and dendrites from other neurons. Bar is 10 µm. (**g**) Summary of RhoA activation levels at dendrite tips in control (left) and BAI1 Kd (right) neurons through time. RhoA activation levels in stable neurons were also stable and assigned a value of 0 throughout the time course. All data were scaled to the average value of FRET in control somata on 16 DIV for the appropriate repeat of the experiment. (**h**) Same as panel g, but for dendrite branch points. (**i**) RhoA activation levels from all dendrites (stable, retracting, and extending) were pooled and plotted against day-long changes in dendritic length in control (black) and BAI1 Kd (green) neurons and linear fits were applied as shown. The inset to this panel shows the slopes extracted from these linear fits. Total number of dendrites represented in panels e-h, 922 for control (from 50 neurons) and 821 for BAI1 Kd (from 38 neurons) (N = 5). Data are represented as mean ± s.e.m., except for the inset to panel i, which is shown ± 95% confidence intervals. (***$p<0.0001$, **$p<0.01$, *$p<0.05$; §$p<1E-6$ vs. control retracting) Detailed statistics are found in *Figure 3—source data 1*.

DOI: https://doi.org/10.7554/eLife.47566.017

The following source data and figure supplements are available for figure 3:

**Source data 1.** Statistical summary for *Figure 3*: ANOVA and key Tukey *post-hoc* tests and N and n values for *Figure 3b,c,g,h*.
DOI: https://doi.org/10.7554/eLife.47566.028

**Source data 2.** Individual values for RhoA activation and related values over fixed and longitudinal time courses (*Figure 3b,c,g–i*).
DOI: https://doi.org/10.7554/eLife.47566.024

**Source data 3.** Individual measurements for long-term Rho-GTPase measurements in Cos-7 cells (*Figure 3—figure supplement 1b,d,f,h,j,k*).
DOI: https://doi.org/10.7554/eLife.47566.025

**Source data 4.** Individual measurements of Rac1 and correlations between RhoA activation and growth (*Figure 3—figure supplement 2b,c,e–g*).
DOI: https://doi.org/10.7554/eLife.47566.026

**Source data 5.** Individual measurements of arbor parameters in control and RhoA KO mouse neurons (*Figure 3—figure supplement 3b–e*).
DOI: https://doi.org/10.7554/eLife.47566.027

**Figure supplement 1.** Long-term measurements of Rho-GTPase activation.
DOI: https://doi.org/10.7554/eLife.47566.018

**Figure supplement 1—source data 1.** Statistical summary for *Figure 3—figure supplement 1*: ANOVA and key Tukey *post-hoc* tests and N and n values for *Figure 3—figure supplement 1b,d,f,h,j*.
DOI: https://doi.org/10.7554/eLife.47566.019

**Figure supplement 2.** Effect of BAI1 Kd on dendritic Rac1 activation through development and lack of robust correlation of some RhoA parameters with dendrite growth.
DOI: https://doi.org/10.7554/eLife.47566.020

**Figure supplement 2—source data 1.** Statistical summary for *Figure 3—figure supplement 2*: ANOVA and key Tukey *post-hoc* tests and N and n values for *Figure 3—figure supplement 2b,c*.
DOI: https://doi.org/10.7554/eLife.47566.021

**Figure supplement 3.** Loss of RhoA leads to late overgrowth of dendrites in mouse hippocampal neurons.
DOI: https://doi.org/10.7554/eLife.47566.022

**Figure supplement 3—source data 1.** Statistical summary for *Figure 3—figure supplement 3*: ANOVA and key Tukey *post-hoc* tests and N and n values for *Figure 3—figure supplement 3b–e*.
DOI: https://doi.org/10.7554/eLife.47566.023

reported, BAI1 Kd caused decreases in spine density (*Figure 4—figure supplement 2a,b*) and maximum diameter (*Figure 4—figure supplement 2a,e*), but increases in filopod density (*Figure 4—figure supplement 2a,c*) and spine length (*Figure 4—figure supplement 2a,d*). Unlike dendritic arbor morphology, narciclasine did not rescue the spine density defects (*Figure 4—figure supplement 2a, b*), though it did block increases in spine length (*Figure 4—figure supplement 2a,d*) and decreased spine maximum diameter by itself (*Figure 4—figure supplement 2a,e*), consistent with known effects of RhoA on spine morphology (*Tolias et al., 2011*).

We next turned to the BAI1 gain-of-function phenotype. RhoA-dependent kinases, especially ROCK1/2, are implicated in dendrite growth arrest (*Li et al., 2015*; *Sin et al., 2002*; *Nakayama et al., 2000*), so we tested the ability of the ROCK inhibitor Y-27632 to rescue dendritic arbors from the contraction caused by BAI1 OX (*Figure 2*). As with narciclasine, we used a low dose (2 µM) of Y-27632 that had no effect on control hippocampal neurons treated on 17 DIV and assayed

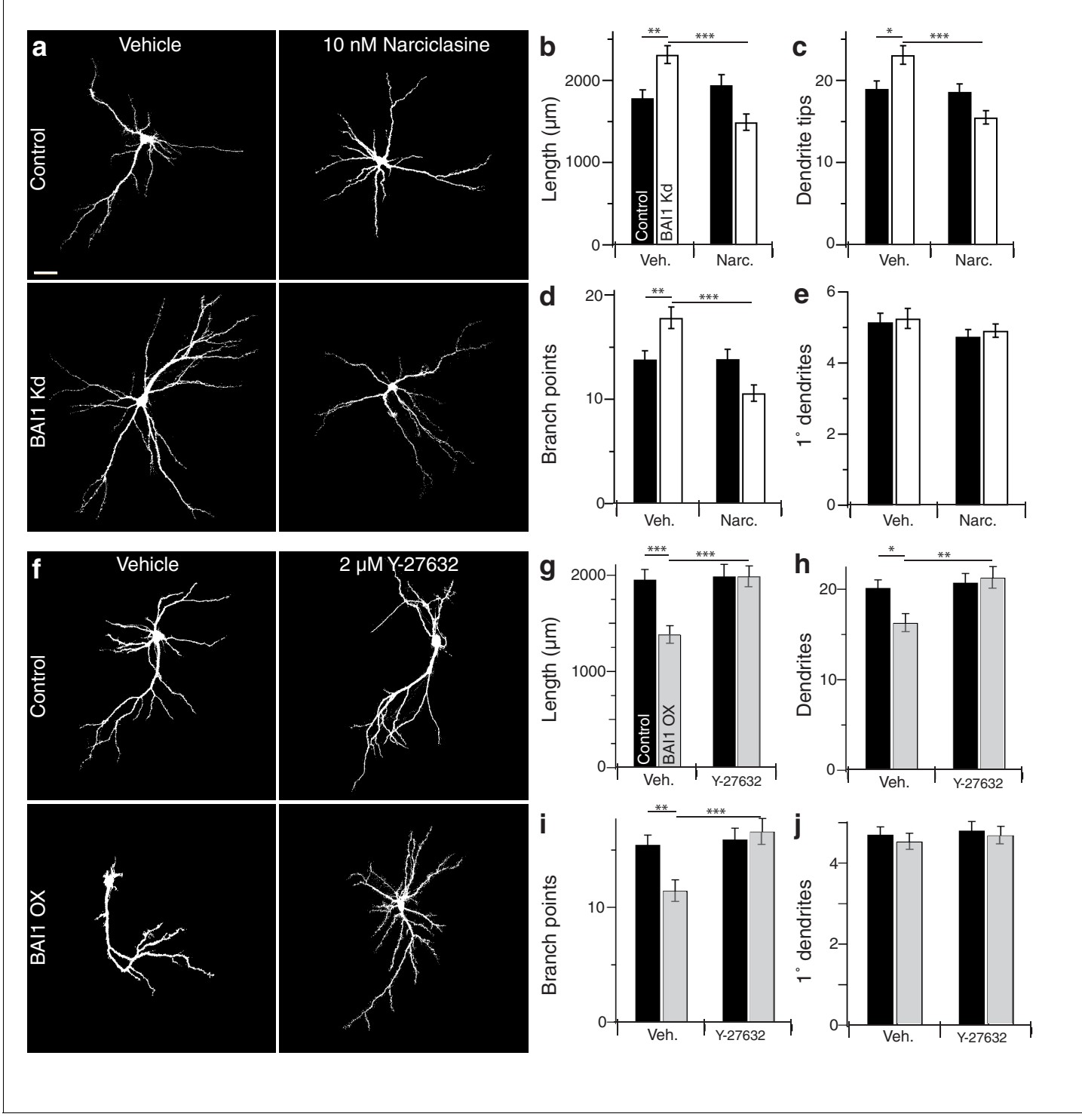

**Figure 4.** BAI1 manipulations are rescued by compensatory changes in RhoA signaling. (a) Representative somatodendritic images of hippocampal neurons transfected with EGFP alone (control) or with shRNA against *Adgrb1* (BAI1 Kd) treated with vehicle or 10 nM narciclasine at 17 DIV and fixed at 21 DIV. Images are masked to remove axons and dendrites from other neurons. Bar is 20 µm. (b) Summary data for dendrite length. (c) Summary data for dendrite number. (d) Summary data for branches. (e) Summary data for primary dendrites. Total number of neurons: 105 per condition (N = 3). (f) Representative somatodendritic images of hippocampal neurons transfected with EGFP alone (control) or with BAI1 (BAI1 OX) treated with vehicle or 2 µM Y-26732 at 17 DIV, fixed at 21 DIV and masked as in panel a. g) Summary data for dendrite length. (h) Summary data for dendrite number. (i) Summary data for branch number. (j) Summary data for primary dendrite number. Total number of neurons: 47 per condition (N = 4). Data are represented ± s.e.m. (***p<0.0001, **p<0.01, *p<0.05) Detailed statistics are found in *Figure 4—source data 1*.

*Figure 4 continued on next page*

*Figure 4 continued*

DOI: https://doi.org/10.7554/eLife.47566.029

The following source data and figure supplements are available for figure 4:

**Source data 1.** Statistical summary for *Figure 4*: ANOVA and key Tukey *post-hoc* tests and N and n values for *Figure 4b–e,g–j*.
DOI: https://doi.org/10.7554/eLife.47566.037
**Source data 2.** Individual measurements of arbor parameters for neurons treated with narciclasine and Y-27632 (*Figure 4b–e,g–j*).
DOI: https://doi.org/10.7554/eLife.47566.034
**Source data 3.** Individual measurements of narciclasine controls and Sholl analyses in narciclasine- and Y-27632-treated neurons (*Figure 4—figure supplement 1b–g*).
DOI: https://doi.org/10.7554/eLife.47566.035
**Source data 4.** Individual measurements of spine parameters in neurons treated with narciclasine (*Figure 4—figure supplement 1b–e*).
DOI: https://doi.org/10.7554/eLife.47566.036
**Figure supplement 1.** Verification of narciclasine function in neurons and Sholl analyses with RhoA manipulation.
DOI: https://doi.org/10.7554/eLife.47566.030
**Figure supplement 1—source data 1.** Statistical summary for *Figure 4—figure supplement 1*: ANOVA and key Tukey *post-hoc* tests and N and n values for *Figure 4—figure supplement 1b–g*.
DOI: https://doi.org/10.7554/eLife.47566.031
**Figure supplement 2.** Effect of narciclasine on dendritic spines.
DOI: https://doi.org/10.7554/eLife.47566.032
**Figure supplement 2—source data 1.** Statistical summary for *Figure 4—figure supplement 2*: ANOVA and key Tukey *post-hoc* tests and N and n values for *Figure 4—figure supplement 2b–e*.
DOI: https://doi.org/10.7554/eLife.47566.033

on 21 DIV (*Figure 4f*). As above, BAI1 OX led to lower total dendrite length (*Figure 4f,g*), fewer dendrites (*Figure 4f,h*), fewer branch points (*Figure 4f,i*), and contracted Sholl curves (*Figure 4—figure supplement 1f,g*). All of these effects were prevented by Y-27632 (*Figure 4–i*, *Figure 4—figure supplement 1f–g*). Y-27632 did not affect the number of primary dendrites (*Figure 4f,j*). Thus, modulating RhoA signaling with narciclasine (RhoA activation) or Y-27632 (ROCK inhibition) rescues dendritic arbors from the effects of altered BAI1 signals, suggesting a role for RhoA signaling downstream of BAI1.

## No previously described signaling function of BAI1 mediates RhoA activation or dendrite growth arrest

A growing body of research attests to the complexity, modularity, and variable outcomes of BAI1 signaling (*Duman et al., 2016*). *Figure 5—figure supplement 1a* shows BAI1's domain structure and summarizes the signaling pathways traced to specific BAI1 domains to date. There are four known intracellular signaling motifs in BAI1, and we tested each for their involvement in arbor growth restriction using a molecular replacement strategy. BAI1 has been linked to Rho-family GTPase guanine-nucleotide exchange factors (Rho-GEFs), which activate Rho-family small GTPases, through three motifs: (i) the PDZ domain-binding TEV motif at the extreme C-terminus of the protein, which recruits the Rac1-GEF complex Tiam1/Par3 in neurons (*Duman et al., 2013*), (ii) a helical domain that recruits and activates the Rac1-GEF complex ELMO/DOCK180 in a ligand-dependent manner (*Park et al., 2007*), and (iii) the GPCR domain, which couples to $G\alpha_{12/13}$ and activates RhoA (*Stephenson et al., 2013*). Deletion or mutation of each of these domains resulted in well-expressed mutants that are surface-localized (*Figure 5—figure supplement 2a,b* and *Tu et al., 2018*) and capable of rescuing BAI1 Kd-induced increases in dendrite length, number, and branches as effectively as wild-type BAI1 (*Figure 5a–c,f–h*). It is especially interesting that the ΔTEV BAI1 mutant rescued arbor growth arrest, as it does not rescue spine development (*Duman et al., 2013*), arguing against BAI1's effect on arbors arising as a secondary effect from its effects on synaptogenesis. To further address this point, we demonstrated that suppression of activity did not cause arbor overgrowth in control hippocampal neurons, nor did increasing activity prevent overgrowth in BAI1 Kd neurons (*Figure 5—figure supplement 1h,i*). We also deleted a proline-rich region that links BAI1 to the scaffold IRSp53 (*Oda et al., 1999*), and found that this mutant also rescued BAI1 Kd-mediated dendrite overgrowth (*Figure 5d,f–h*). Neither BAI1 Kd nor any of the mutants affected primary dendrite number (*Figure 5—figure supplement 1b*), and all mutants were expressed at levels

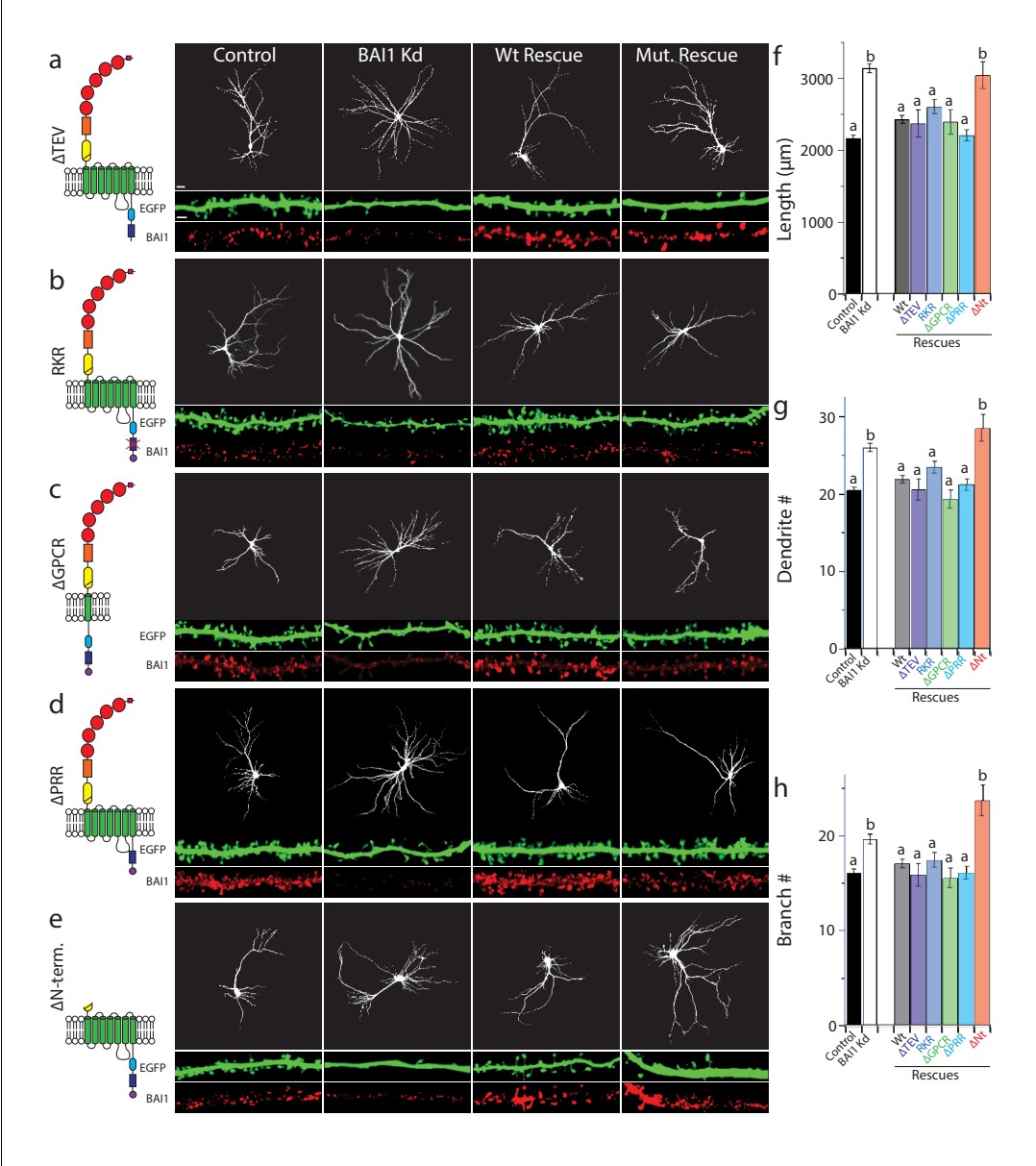

**Figure 5.** BAI1 requires its extracellular segment to inhibit dendrite growth, but does not employ any known intracellular signaling modality to do so. a through e Representative images of neurons expressing EGFP and control vector (Control), shRNA against *Adgrb1* (BAI1 Kd), or shRNAs against BAI1 plus RNAi-resistant BAI1 (Wt rescue) or an RNAi-resistant BAI1 mutant (Mut. rescue). All images are masked to remove axons and dendrites from other neurons. The mutant used for rescue in each series is shown in cartoon form on the left. Mutants used included a deletion of the PDZ-binding core sequence (a), an RKR→AAA mutation in the helical domain that prevents ELMO1 binding (b), deletion of all but the first transmembrane domain of the GPCR domain (c), deletion of the proline-rich region (d), and deletion of the entire N-terminal extracellular segment between the signal sequence and the native autoproteolysis site (e). Representative EGFP images of the entire arbor are on top (bar is 20 μm), while EGFP and BAI1 immunochemistry images reconstructed from 3D confocal stacks of secondary dendrites taken from the same experiments are shown immediately under these (bar is 2.5 μm). (f) Summary data for dendrite length. (g) Summary data for dendrite number. (h) Summary data for branch number. Total number of neurons: 314 for control neurons (N = 24), 288 for BAI1 Kd neurons (N = 24), 235 for wt rescue neurons (N = 24), 29 for ΔTEV rescue neurons (N = 3), 86 for RKR rescue neurons (N = 9), 39 for ΔGPCR neurons (N = 3), 72 for ΔPRR neurons (N = 3), and 37 for (N = 3) for ΔNterm. rescues. Data are represented ± s. e.m. ([a]P >0.05 within group *a*, but p<0.05 vs. group *b*; [b]P >0.05 within group *b*, but p<0.05 vs. group *a*.) Detailed statistics are found in *Figure 5— source data 1*.

DOI: https://doi.org/10.7554/eLife.47566.038

The following source data and figure supplements are available for figure 5:

**Source data 1.** Statistical summary for *Figure 5*: ANOVA and key Tukey *post-hoc* tests and N and n values for *Figure 5f–h*.

*Figure 5 continued*

DOI: https://doi.org/10.7554/eLife.47566.044

**Source data 2.** Individual measurements of dendritic parameters in control neurons and those with molecular replacements with BAI1 mutants (*Figure 5*).
DOI: https://doi.org/10.7554/eLife.47566.042

**Source data 3.** Individual measurements of primary dendrites in molecular replacement experiments and dendrite arbor lengths when neuronal activity is modulated (*Figure 5—figure supplement 1b,i*).
DOI: https://doi.org/10.7554/eLife.47566.043

**Figure supplement 1.** Signaling pathways downstream of BAI1; 1° dendrites in molecular replacement studies; and effect of activity on arbor form.
DOI: https://doi.org/10.7554/eLife.47566.039

**Figure supplement 1—source data 1.** Statistical summary for *Figure 5—figure supplement 1* and summary of ANOVA and key Tukey *post-hoc* tests and N and n values for *Figure 5—figure supplement 1b, i*.
DOI: https://doi.org/10.7554/eLife.47566.040

**Figure supplement 2.** Surface labeling of BAI1 and mutants.
DOI: https://doi.org/10.7554/eLife.47566.041

comparable to wild-type BAI1 (*Figure 5a–e*, *Figure 5—figure supplement 1c–f*). In sum, no previously reported BAI1 signaling modality is necessary for restricting dendrite growth.

Additionally, we deleted the N-terminal segment of BAI1 between the signal sequence and GPS motif within its GAIN domain (see *Figure 5* legend and discussion), mimicking an archetypal A-GPCR intramolecular cleavage (*Hamann et al., 2015*). This segment contains a number of motifs and domains that have putative or demonstrated ligand-binding functions (*Hamann et al., 2015*; *Langenhan et al., 2013*; *Yona et al., 2008*). BAI1-ΔN-term. traffics to the cell surface, expresses well, and localizes to dendrites (*Tu et al., 2018* and *Figure 5e*, *Figure 5—figure supplement 1g*). However, unlike the other mutants, it did not rescue dendritic overgrowth in BAI1 Kd neurons (*Figure 5e,f–h*). Thus, extracellular ligand binding is likely required for BAI1-mediated arbor growth arrest.

Our panel of mutants (*Figure 5*) allowed us to further verify the relationship between dendrite growth and RhoA activation. As shown previously, BAI1 Kd neurons had clearly lower levels of RhoA activation throughout their somatodendritic domains (*Figure 6a*), and especially at dendrite tips (*Figure 6b*). Both dendrite overgrowth and lowered RhoA activation were rescued by the expression of RNAi-resistant BAI1 (*Figure 6a,b*). Further, all of the BAI1 mutants that rescued dendrite overgrowth (ΔTEV, RKR, ΔGPCR, and ΔPRR) also rescued RhoA activation, while BAI1-ΔN-term. failed to rescue both defects (*Figure 6a,b*). Taken together, these results strongly suggest that BAI1 functions through RhoA to mediate dendritic arbor growth arrest.

## Bcr links BAI1 to RhoA and dendritic growth arrest

How does BAI1 regulate RhoA in the context of dendritic growth arrest? We performed a small screen in which we immobilized BAI1's intracellular C-terminal tail on glutathione resin and assayed its interaction with dendritic RhoA regulatory proteins. In so doing, we identified the related proteins breakpoint cluster region (Bcr) and activated Bcr-related (Abr) (*Figure 7—figure supplement 1a*), both of which restrict dendritic growth via inhibition of Rac1, but also have putative RhoA-GEF domains (*Um et al., 2014*). We confirmed these interactions by immunoprecipitation from hippocampal neurons. Interestingly, we detected little or no interaction between BAI1 and Bcr before DIV 17, but the interaction was readily detected at the time of dendritic growth arrest (*Figure 7a–c*). No interaction was detected between BAI1 and Abr (*Figure 7a,b*).

If BAI1 is required to activate the RhoA-GEF activity of Bcr and/or Abr, then it is possible that their overexpression might compensate for BAI1 loss. We tested this in hippocampal neurons. BAI1 Kd led to longer, more complex arbors as before, and both Bcr and Abr overexpression led to contracted arbors (*Figure 7d–f*) as previously reported (*Park et al., 2012*). However, Bcr OX balanced BAI1 Kd, giving rise to arbors of normal length and complexity while Abr OX did not (*Figure 7d–f*). This difference could not be explained by differences in Rac1 inhibitory activity (*Figure 7—figure supplement 1b,c*). Moreover, the effectiveness of Bcr in causing arbor contraction, assayed by the final arbor length as a function of Bcr OX level, was blunted by BAI1 loss (*Figure 7g–i*). This supports the idea that BAI1 activates Bcr RhoA-GEF activity in neurons. No such functional interaction with

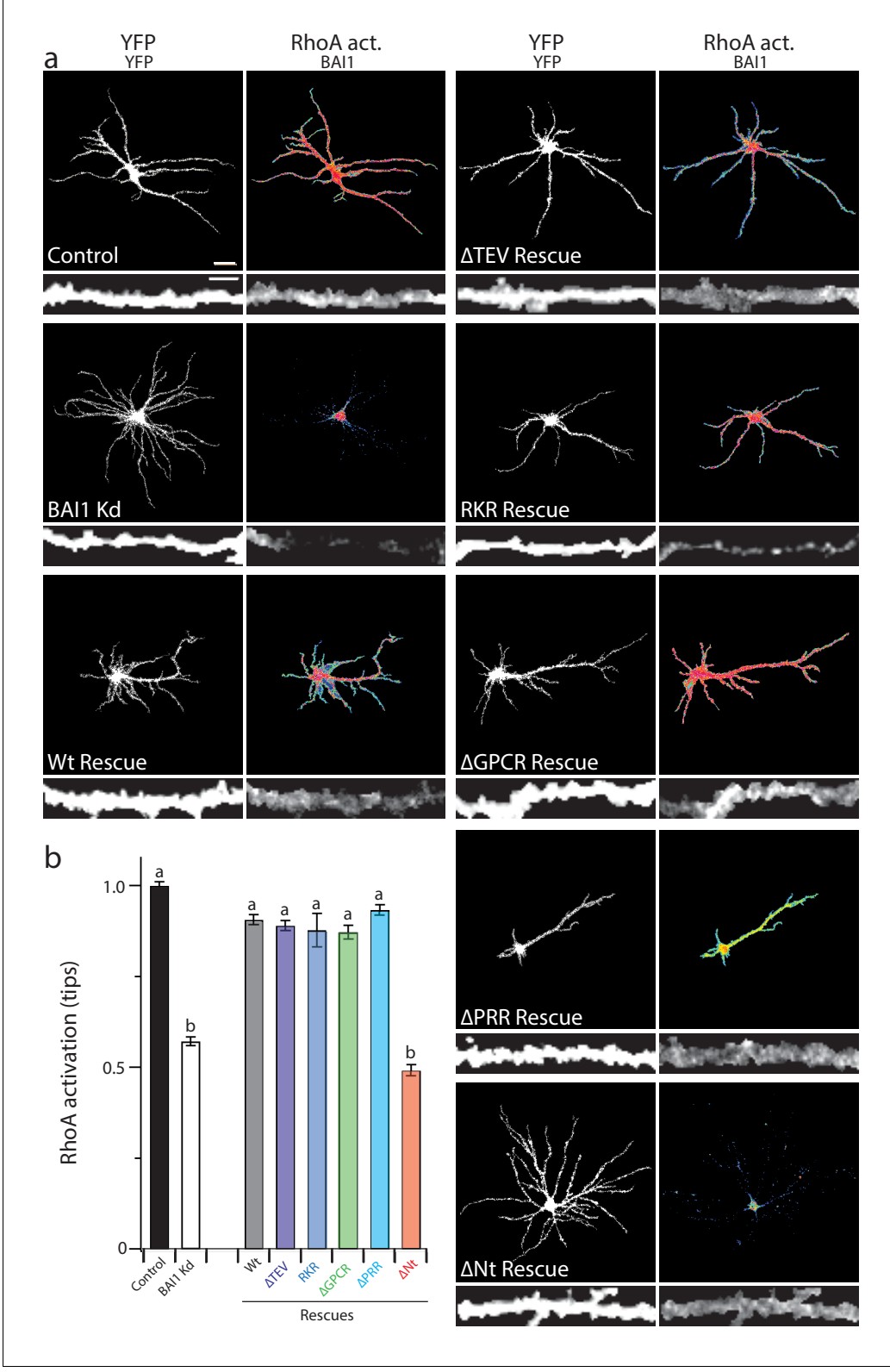

**Figure 6.** Proper RhoA activation is required for rescue of the BAI1 Kd-mediated dendrite overgrowth. (a) Representative images of hippocampal neurons expressing the RhoA activation reporter RhoA-FLARE, along with control vector (control), shRNA against *Adgrb1* (BAI1 Kd), or shRNA against BAI1 and RNAi-resistant BAI1 (Wt rescue) or the indicated RNAi-resistant mutants (see *Figure 6*). The 4-panel cluster for each condition consists of

*Figure 6 continued on next page*

*Figure 6 continued*
an image of the complete somatodendritic domain obtained by direct excitation of the acceptor (YFP images, upper left; bar is 20 µm), a map of RhoA activation color-coded according the scale in *Figure 3* (RhoA Act., upper right), and digitally magnified stretches of dendrite showing morphology obtained by direct excitation of the acceptor (YFP, lower left; bar is 5 µm), and BAI1 immunostaining (BAI1, lower right). Images are masked to remove axons and dendrites from other neurons. (**b**) Summary data of RhoA activation at dendritic tips. Total number of dendrite tips: 670 for controls, 495 for BAI1 Kd, 630 for wt rescues, 640 for ΔTEV rescues, 60 for RKR rescues, 40 for ΔGPCR rescues, 635 for ΔPRR rescues, and 55 for ΔNterm. rescues (N = 5). Data are represented ± s.e.m. ([a]P >0.05 within group *a*, but p<0.05 vs. group *b*; [b]P >0.05 within group *b*, but p<0.05 vs. group *a*.) Detailed statistics are found in *Figure 6—source data 1*.
DOI: https://doi.org/10.7554/eLife.47566.045
The following source data is available for figure 6:

**Source data 1.** Statistical summary for *Figure 6*: ANOVA and key Tukey *post-hoc* tests and N and n values for (*Figure 6b*).
DOI: https://doi.org/10.7554/eLife.47566.047
**Source data 2.** Individual measurements of RhoA activation at dendrite tips in molecular replacement experiments (*Figure 6b*).
DOI: https://doi.org/10.7554/eLife.47566.046

Abr was detected (*Figure 7—figure supplement 1d,e*). Thus, both the biochemical and functional data support a role for Bcr in BAI1 signaling rather than Abr.

Despite sequence predictions that Bcr is a RhoA-GEF, there is little evidence that it functions in this capacity outside of the pathological Bcr-Abl fusion (see discussion). We detected RhoA activation in response to Bcr expression in Cos-7 cells (*Figure 7j,k*). Moreover, co-expression with BAI1 greatly enhanced Bcr RhoA-GEF activity (*Figure 7j,k*), despite comparable expression of the proteins in the cells (*Figure 7—figure supplement 1f*), suggesting that BAI1 stimulates this activity. Is this Bcr-RhoA-GEF activity required for dendrite growth arrest? We tested this using hippocampal neurons isolated from Bcr$^{-/-}$ mice and wild-type littermates. As previously reported (*Park et al., 2012*), Bcr loss caused dendritic overgrowth that was rescued by expression of wild type Bcr (*Figure 8a–c*). Strikingly, when the rescue was attempted using mutants of Bcr deficient for either Rac1-GAP (as previously reported [*Park et al., 2012*]) or RhoA-GEF activity, neither dendritic length (*Figure 8b*) nor dendrite number (*Figure 8c*) was rescued. These results could not be explained by lower expression of the mutants (*Figure 8a*). To test whether both of these activities are required to complement the loss of BAI1, we performed an experiment analogous to that shown in *Figure 7d–f*, using the Bcr mutants in place of Abr. Overexpression of neither mutant rescued BAI1 Kd-mediated dendrite overgrowth in rat hippocampal neurons assayed by length (*Figure 8d,e*) or dendrite number (*Figure 8d,f*). These results suggest that BAI1-orchestrated dendrite growth arrest requires RhoA activation by Bcr downstream of BAI1 in concert with the Rac1-GAP activity of Bcr. This suggests that dendrite growth arrest requires a great change in the signaling environment within dendrites akin to a state change.

## Discussion

We show here that lack of BAI1 in hippocampal neurons leads to longer, more complex, less stable dendritic arbors in culture and in vivo, whereas BAI1 overexpression leads to dendrite retraction. Strikingly, both defects manifest only late in development when dendritic arbors are transitioning from growth to maintenance, despite BAI1 having been manipulated well before this time. None of the previously known BAI1 signaling mechanisms accounts for dendritic growth arrest; rather, BAI1 couples to the small GTPase RhoA through Bcr independently of its GPCR moiety. RhoA is known to inhibit dendritic growth (*Sin et al., 2002*; *Wong et al., 2000*; *Nakayama et al., 2000*; *Lee et al., 2000*; *Li et al., 2000*; *Ruchhoeft et al., 1999*), but the dynamics of RhoA activation throughout dendritic development are not well understood (see below). BAI1 loss postpones and attenuates the previously unreported peak of RhoA activation coincident with dendrite growth arrest. Moreover, BAI1 loss globally suppresses RhoA activity and uncouples dendritic growth from RhoA activation at the tips, which seems to drive dendritic behavior late in development. This Bcr-dependent activation

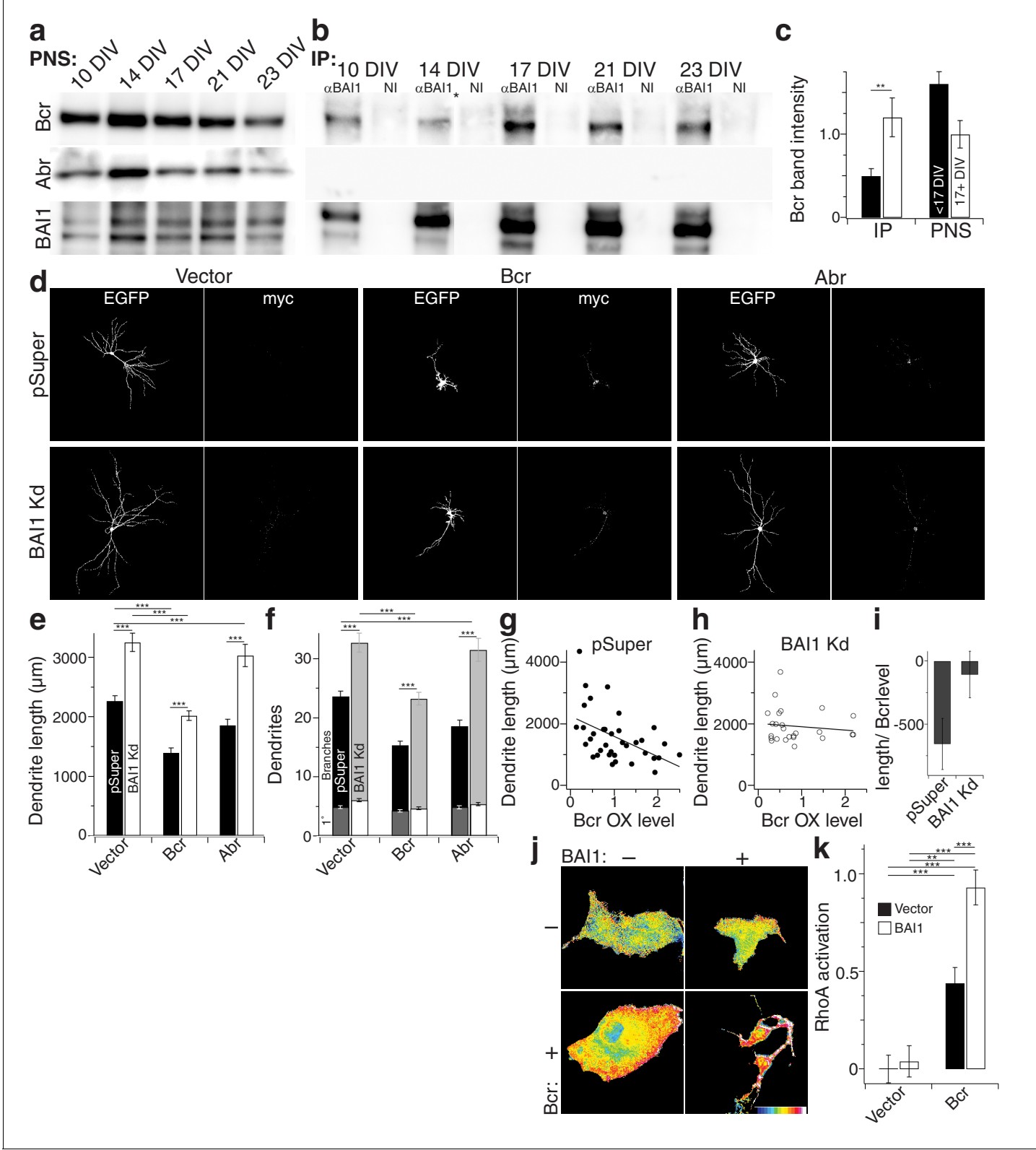

**Figure 7.** BAI1 signals to RhoA via Bcr. (a) Representative Western blots of Bcr, Abr, and BAI1 from primary hippocampal postnuclear supernatants (PNS) prepared at the indicated times from the same preparation. (b) Representative immunoprecipitations from the PNSs in panel a using an antibody against the C-terminus of BAI1 or non-immune serum (NI). A lane was removed from this blot due to a loading irregularity, and this is marked by an asterisk (*). (c) Summary data of Bcr band intensity from the experiment in panels a and b. Total number of preparations: N = 5. (d) Representative

*Figure 7 continued on next page*

*Figure 7 continued*

images of hippocampal neurons expressing EGFP and control shRNA vector (pSuper) or shRNA against *Adgrb1* (BAI1 Kd) and control myc-vector (vector), myc-tagged Bcr, or myc-tagged Abr. (e) Summary data of dendrite length in the experiment in panel d. (f) Summary data of dendrite and branch number in the experiment in panel d. (g) Scatter plot of dendrite length vs. quantified myc level in Vector pSuper neurons including a linear fit of the data. (h) Same as panel g, except for Bcr BAI1 Kd neurons. (i) Slopes of the lines in panels g and h, shown with 95% confidence intervals. (j) Representative images of Cos-7 cells expressing Raichu-RhoA and the indicated combinations of BAI1 and Bcr. RhoA activation is shown in the FRET images according to the color code in the lower right corner (low values on the left, high on the right). (k) Summary data of RhoA activation at the plasma membrane. Total number of cells: 452 for all conditions (N = 10). Data are represented ± s.e.m. except for panel i. (\*\*\*p<0.0001, \*\*p<0.01, \*p<0.05) Detailed statistics are found in *Figure 7—source data 1*.

DOI: https://doi.org/10.7554/eLife.47566.048

The following source data and figure supplements are available for figure 7:

**Source data 1.** Statistical summary for *Figure 7*: ANOVA and key Tukey *post-hoc* tests and N and n values for *Figure 7c,e,f,k*.

DOI: https://doi.org/10.7554/eLife.47566.053

**Source data 2.** Individual measurements of Western blot bands, arbor parameters with Bcr and Abr, and Cos-7 RhoA activation (*Figure 7c,e,f,k*).

DOI: https://doi.org/10.7554/eLife.47566.051

**Source data 3.** Individual measurements of Rac1 activation in control, Abr-, and Bcr-expressing neurons (*Figure 7—figure supplement 1c*).

DOI: https://doi.org/10.7554/eLife.47566.052

**Figure supplement 1.** Bcr functionally interacts with BAI1, while Abr does not.

DOI: https://doi.org/10.7554/eLife.47566.049

**Figure supplement 1—source data 1.** Statistical summary for *Figure 7—figure supplement 1*: ANOVA and key Tukey *post-hoc* tests and N and n values for *Figure 7—figure supplement 1c*.

DOI: https://doi.org/10.7554/eLife.47566.050

of RhoA, however is not sufficient to cause growth arrest but also requires Bcr-mediated inhibition of the complementary small GTPase Rac1. These results strongly implicate a BAI1-Bcr-RhoA pathway in regulating dendritic volume in the hippocampus, which is intriguing, especially given the links between BAI1 and ASD (*Michaelson et al., 2012*; *Chahrour et al., 2008*; *Urdinguio et al., 2008*).

We previously reported that BAI1 Kd also causes a marked decrease in spine density in both cultured hippocampal neurons and in vivo in CA1 and cortical pyramidal neurons (*Duman et al., 2013*; *Tu et al., 2018*). That BAI1 Kd leads to a loss of dendritic spines and synapses in hippocampal neurons (*Duman et al., 2013*; *Tu et al., 2018*) could be misconstrued as the explanation for the longer arbors reported here. Specifically, if low synapse density prevents BAI1 Kd neurons from achieving a synaptic 'quota' (i.e. reaching a set level of activity), neurons may compensate by extending their dendrites, possibly by inhibiting RhoA. This resembles the behavior of the *Drosophila* aCC motoneuron, whose arbor expands when deprived of afferent inputs (*Tripodi et al., 2008*). However, our data are inconsistent with this explanation. First, the BAI1•ΔTEV mutant rescues both the dendrite overgrowth (*Figure 5a,f–h*) and the decreased dendritic RhoA activation (*Figure 6*) effects of BAI1 Kd, whereas it does *not* rescue the spine deficits (*Duman et al., 2013* and *Figure 5a*), indicating that arbor and spine development are independently regulated by BAI1. Second, ectopic RhoA activation rescues arbor overgrowth (*Figure 4a–d*), but does not rescue spine loss (*Figure 4—figure supplement 2a,b*), again demonstrating the independence of these effects. Finally, hippocampal neurons do not overgrow when either NMDA receptors or voltage-gated Na$^+$ channels are blocked, nor does increasing neural activity cause retraction of BAI1 Kd neurons (*Figure 5—figure supplement 1h,i*). Thus, we can affirm that the effect of BAI1 on dendritic arbors is not a side effect of synapse loss. Is this result unexpected? Recently, two independent groups ablated presynaptic glutamate in the hippocampus using different strategies, and both showed relatively normal dendritic arbor length with no evidence of overgrowth in either CA1 pyramidal neurons or dentate granule cells at P21 (*Sando et al., 2017*; *Sigler et al., 2017*). Moreover, dendritic hypertrophy is independent of synapse dysfunction in the PTEN mouse model of ASD (*Sperow et al., 2012*). Thus, while neural activity can greatly affect the length, form, and/or orientation of some neurons' dendrites (*Lefebvre et al., 2015*; *Dong et al., 2015*), this does not appear to be the case for hippocampal neurons. Therefore, we conclude that BAI1's role in restricting arbor growth is independent of its effect on promoting dendritic spinogenesis/synaptogenesis, and is mediated by the activation of RhoA by Bcr.

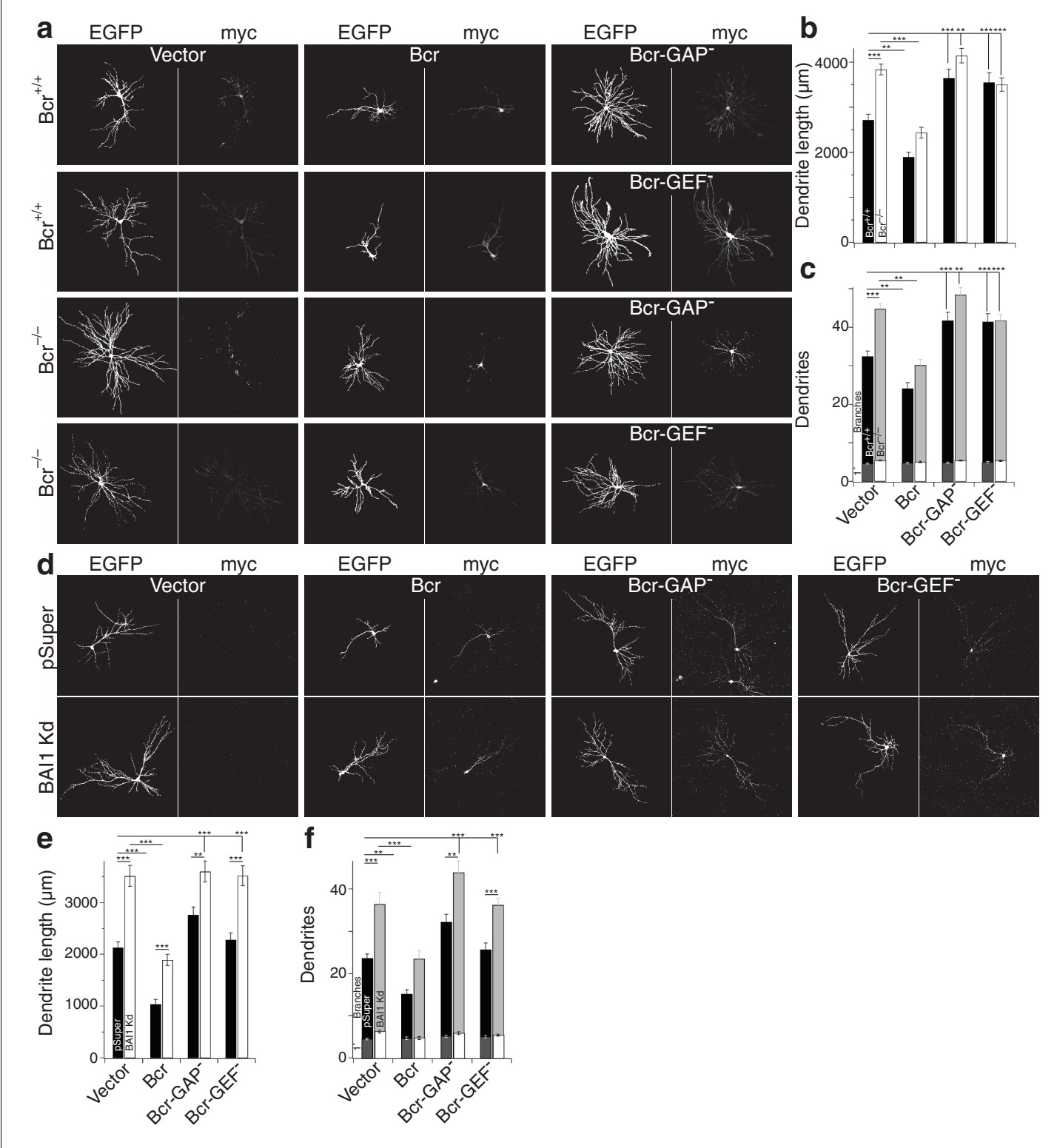

**Figure 8.** Both the RhoA-GEF and Rac1-GAP functions of Bcr are required for dendrite growth arrest. (a) Representative images of mouse neurons with the indicated genotype expressing EGFP, pSuper or shRNA against *Adgrb1,* and empty vector (Vector), wild-type Bcr (Bcr), GAP-dead Bcr (Bcr-GAP⁻), or GEF-dead Bcr (Bcr-GEF⁻). All cells within a row were cultured from the same individual mouse. (b) Summary data of dendrite length. (c) Summary data for dendrite numbers and branches. Total number of neurons: 147 for Bcr^{+/+} Vector, 82 for Bcr^{+/+} Bcr, 69 Bcr^{+/+} Bcr-GAP⁻, 69 Bcr^{+/+}-GEF⁻, 177 for Bc^{r−/−} Vector, 99 for Bcr^{−/−} Bcr, 75 Bcr^{−/−} Bcr-GAP⁻, and 109 for Bcr^{−/−} Bcr-GEF⁻ (N = 6–12). (d) Representative images of rat neurons expressing EGFP

*Figure 8 continued on next page*

*Figure 8 continued*

and pSuper or BAI1 Kd and the vectors as described in panel a. (e) Summary data of dendrite length. (f) Summary data of dendrite number and branches. Total number of neurons: 52 for pSuper Vector, 52 for pSuper Bcr, 37 pSuper Bcr-GAP⁻, 60 for pSuper Bcr-GEF⁻, 34 for BAI1 Kd Vector, 43 BAI1 Kd Bcr, 33 BAI1 Kd Bcr-GAP⁻, and 47 BAI1 Kd Bcr-GEF⁻ (N = 5). Data are represented ± s.e.m. (***p<0.0001, **p<0.01, *p<0.05) Detailed statistics are found sin *Figure 8—source data 1*.

DOI: https://doi.org/10.7554/eLife.47566.054

The following source data is available for figure 8:

**Source data 1.** Statistical summary for *Figure 8*: ANOVA and key Tukey *post-hoc* tests and N and n values for *Figure 8b,c,e,f*.
DOI: https://doi.org/10.7554/eLife.47566.056
**Source data 2.** Individual arbor parameters for mouse and rat neurons expressing different forms of Bcr *Figure 8B,C,E,F*.
DOI: https://doi.org/10.7554/eLife.47566.055

A recent study characterizing global BAI1 KO mice reported no change in dendritic arbor length or spine density as a result of BAI1 loss, though no quantitative dendritic data were presented (*Zhu et al., 2015*). These animals have diminished hippocampus-based learning, lowered threshold for LTP, LTP in response to LTD-provoking stimuli, and decreased PSD thickness (*Zhu et al., 2015*). BAI1 KO mice also have decreased levels of the postsynaptic scaffold PSD95, which we also reported in our sparse Kd models (*Duman et al., 2013*; *Tu et al., 2018*), and BAI1 stabilizes PSD95 by binding to and inhibiting the E3 ubiquitin ligase MDM2 (*Zhu et al., 2015*). While these results of Zhu et al. may seem to be at odds with ours, this is not necessarily the case. First, Zhu et al. did not account for potential compensation by BAI2 and/or BAI3 (*Shiratsuchi et al., 1997*), while our sparse, acute manipulations of neurons did not elicit any such compensation (*Figure 1—figure supplement 1*). Further, BAI1 may assert its effects by conferring an advantage to synapses and/or a disadvantage to dendrites that express it. If this were the case, then global loss of BAI1 would even the playing field and the effects of its loss might be greatly ameliorated. A similar situation may exist for the ASD-associated adhesion protein neuroligin-1 (NL1). While NL1 plays a crucial role in synaptic function, global NL1 KO is associated with essentially no effect on spine or synapse density (*Kwon et al., 2012*). However, if NL1 is removed from a sparse population of neurons, these neurons exhibit dramatically lower spine and synapse densities (*Kwon et al., 2012*); conversely, if NL1 is overexpressed sparsely, these neurons display higher spine densities and larger spines (*Dahlhaus et al., 2010*). Interestingly, we recently demonstrated a functional interaction between BAI1 and NL1 in hippocampal neurons (*Tu et al., 2018*). Based on these considerations, we submit that BAI1 confers an advantage to synapses and a disadvantage to dendritic branches that express it during development. These effects are apparent when BAI1 is overexpressed or knocked down on a background of neurons normally expressing BAI1 in culture or in vivo, but that global removal of BAI1 removes this competitive advantage/disadvantage and makes the effects of BAI1 much less evident.

Bcr possesses multiple functional domains, but its most intriguing feature may be that it possesses a Rac1-GAP domain and a RhoA-GEF domain (*Park et al., 2012*; *Rochelle et al., 2013*; *Tala et al., 2013*). Bcr was previously shown to restrict dendrite development and spinogenesis/synaptogenesis through its Rac1-GAP functionality (*Um et al., 2014*; *Park et al., 2012*), so the requirement for RhoA-GEF activity was unexpected. This activity of Bcr has mostly been described in the context of the p210^bcr-abl fusion from the Philadelphia chromosome that is associated with virtually all cases of chronic myeloid leukemia, and activates RhoA and various downstream targets thereof (*Rochelle et al., 2013*; *Tala et al., 2013*). A physiological role for Bcr's RhoA-GEF activity has been suggested in keratinocyte differentiation, though there is no obvious role for its Rac1-GAP activity in this system (*Dubash et al., 2013*). Our results indicate that Bcr must *both* activate an inhibitor of arborization (RhoA) *and* inhibit an activator of arborization (Rac1) in order to precipitate arbor growth arrest (*Figure 8*). This suggests that the arrest signal is very high in magnitude, and perhaps requires multiple inputs.

## BAI1, RhoA, and dendrite growth

Like all small GTPases, RhoA cycles between a GDP-bound inactive form and a GTP-bound active form (*Evers et al., 2000*; *Ridley, 2006*). Activated RhoA generally mediates inhibitory/repulsive

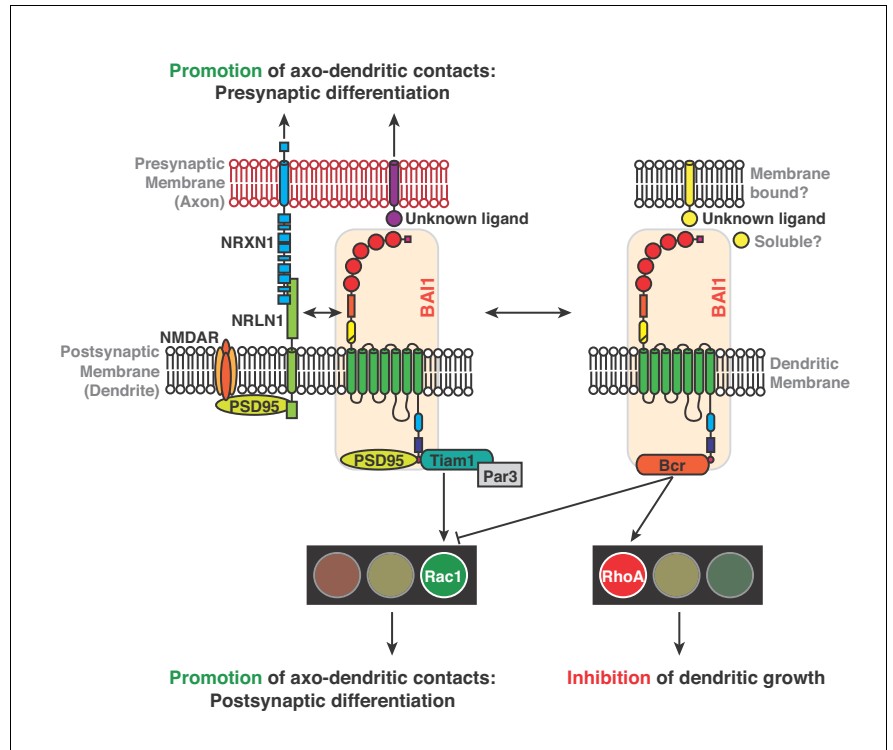

**Figure 9.** Mechanisms for BAI1's action promoting axo-dendritic contacts yet inhibiting dendro-dendritic contacts. Previous work revealed that BAI1 interacts with PSD95 (**Zhu et al., 2015**) and Tiam1/Par3 by which it activates Rac1, promoting synaptogenesis and spinogenesis (**Duman et al., 2013**). Moreover, BAI1 interacts with the synaptic organizer neuroligin 1 (NRLN1) and is required for NRLN1's ability to promote synaptogenesis (**Tu et al., 2018**) and directs presynaptic differentiation through an unknown mechanism (**Tu et al., 2018**). NRLN1 also interacts with neurexin-1 (NRXN1) and with NMDA receptors via PSD95 (**Mondin et al., 2011**). This study reveals an interaction between BAI1 and Bcr that promotes RhoA activation and restricts overall dendrite growth. Bcr also inhibits Rac1, which opposes RhoA activation in promoting dendrite growth.
DOI: https://doi.org/10.7554/eLife.47566.057

processes in the nervous system (**Redmond and Ghosh, 2001**). While RhoA is widely accepted as a physiological antagonist of dendrite growth in a variety of model systems, the studies on which this notion was built primarily employed dominant negative and constitutively active RhoA mutants (**Sin et al., 2002**; **Wong et al., 2000**; **Nakayama et al., 2000**; **Lee et al., 2000**; **Li et al., 2000**; **Ruchhoeft et al., 1999**) or studied the process at a much earlier stage (**Takano et al., 2017**). We now know that overexpression of either type of GTPase mutant has great potential for broad-spectrum effects that cannot definitively be attributed to the target Rho-GTPase (**Schiller, 2006**; **Vega and Ridley, 2007**). However, more recent studies have verified the core finding that RhoA acts to oppose dendrite growth and expanded the contexts within which this might happen: RhoA activation by Arhgef1 opposes dendrite growth in mouse cortex at very early (>P5) stages (**Xiang et al., 2016**); RhoA activation by the neuropeptide orphanin FQ opposes developmental BDNF-stimulated dendrite growth in granule cells of the dentate gyrus (**Alder et al., 2013**); RhoA inhibits the expression of the branch promoter cypin (**Chen and Firestein, 2007**); failure to inactivate RhoA late in development leads to dendrite loss (**Lin et al., 2013**); and UVB injury promotes dendrite growth by decreasing RhoA expression (**Jian et al., 2014**). In all of these studies, RhoA activation was chiefly assayed using techniques that provide no spatial data (**Li et al., 2015**; **Iida et al., 2007**; **Kranenburg et al., 1999**), so very little is known about the developmental dynamics of RhoA activation. We extended these pioneering results by employing a RhoA knockout model (**Mulherkar et al., 2013**) and by using RhoA-specific FRET probes (**Nakamura et al., 2006**; **Pertz et al., 2006**) to measure authentic RhoA effects and spatiotemporal dynamics. In so doing, we elucidated three previously unknown features of dendritic RhoA signaling: (i) RhoA activation peaks

late in development and mediates the transition from dendrite growth to growth arrest; (ii) BAI1 is required to activate RhoA in this context, presumably by transducing some signal and (iii) RhoA activation levels established by 16 DIV determine the ultimate developmental fate of individual persistent dendrites. These results provide a bridge between the expansion of dendritic arbors during development and the requirement for RhoA inhibition to achieve dendrite stability at later time points (*Lin et al., 2013*; *Sfakianos et al., 2007*).

It is worth noting that some part of our conclusions on RhoA function in BAI1 signaling and dendritic growth arrest entails the use of pharmacological agents to manipulate RhoA signaling. We used narciclasine to activate RhoA late in development. Though narciclasine is effective on RhoA at very low concentrations (*Fürst, 2016*; *Lefranc et al., 2009*; *Van Goietsenoven et al., 2013*), it has other targets, including ef1A (*Fürst, 2016*). Narciclasine has marked cytotoxic effects at concentrations at 1 μM, though at 50 nM it has a cytostatic effect, and this is due to its action on the cytoskeleton though RhoA (*Van Goietsenoven et al., 2013*). We observed effective activation of RhoA at 10 nM, and it had no effect on Rac1 at twice the concentration (*Figure 4—figure supplement 1C*). We also acknowledge a lack of specificity of Y-27632 and note that although it inhibits several kinases, it is most effective on those regulated by RhoA (*Davies et al., 2000*). Moreover, since these drugs do not depend on BAI1 and many different signals employ RhoA, it is possible that their effects arise from some parallel pathway. At the concentrations that we use, there was no effect of either drug on the control neurons. We chose these concentrations using the pharmacogenetic strategy of *Stoica et al. (2011)*, in which a subthreshold concentration of drug only acts in synergy with other alterations of the target. Manipulations of BAI1 are the alterations in this case, and the target is RhoA. When considered in the light of the other RhoA-specific techniques presented here, it is thus likely that these drugs mediate their effects by compensating for changes in RhoA signaling caused by BAI1 manipulations.

Even allowing that BAI1 mediates its effects on arbors through RhoA, a number of interesting questions remain. What are the steps that connect BAI1/RhoA to growth arrest? Is it due to the effects of RhoA-regulated kinases on the actin and/or microtubule cytoskeleton? This might seem to be the most likely, yet Rho-GTPases also regulate gene expression (*Bahrami and Drabløs, 2016*) and there are high levels of RhoA in neuronal somata (*Figure 1a*). Rho-GTPases also regulate membrane traffic (*Ridley, 2006*), another process relevant to dendrite growth, and any single effect or combination thereof might contribute to BAI1-dependent arbor sculpting.

In any event, a greater understanding of RhoA signaling in the nervous system is merited, given the role of RhoA in intellectual disabilities (*Nadif Kasri and Van Aelst, 2008*), autism spectrum disorder (*Zunino et al., 2016*), Alzheimer's disease (*Henderson et al., 2016*; *Tsushima et al., 2015*), Parkinson's disease (*Labandeira-Garcia et al., 2015*), and Timothy Syndrome (*Krey et al., 2013*), all of which exhibit altered dendritic arbors (*Kulkarni and Firestein, 2012*; *Krey et al., 2013*). Moreover, since RhoA functions as a second messenger and is in many different signaling pathways, it might be safer, more specific, and ultimately more beneficial to target specific RhoA signaling pathways rather than RhoA generally, even if broad-spectrum inhibition of RhoA can, at times, be beneficial (*Mulherkar et al., 2017*).

## Complexity of BAI1 and A-GPCR signaling

We previously showed that BAI1 promotes excitatory synaptogenesis by recruiting the Rac1 activator complex Par3/Tiam1 (*Duman et al., 2013*). Subsequently, it was demonstrated that BAI1 interacts with the E3 ubiquitin ligase MDM2 to regulate levels of PSD95, an important scaffold in the post-synaptic density, and is required for hippocampal long-term depression (*Zhu et al., 2015*). BAI1 also activates Rac1 via the ELMO1/DOCK180 activator complex, giving rise to phagocytosis (*Park et al., 2007*), couples to RhoA via $G\alpha_{12/13}$ (*Stephenson et al., 2013*), functionally interacts with the synaptic organizer NL1, and signals trans-synaptically to promote presynaptic development (*Tu et al., 2018*). We now show that BAI1 signals to RhoA through a different mechanism, as demonstrated by the rescue of BAI1 Kd phenotypes by BAI1•ΔGPCR (*Figures 5* and *6*). Thus, BAI1 signals through at least five mechanisms, including two pairs of mechanisms that couple to the same second messengers; those mechanisms coupling to Rac1 lead to different cellular outcomes (i.e., synaptogenesis versus engulfment). BAI1's signaling role as we have elucidated it in neurons is summarized in *Figure 9*. These data paint a picture of BAI1 as a signaling integrator that couples to different signaling pathways as appropriate.

How does BAI1 'choose' a downstream pathway? An easy model is that a code in the form of combinations of bound ligands may stipulate the events downstream of BAI1. This is especially plausible when one considers that BAI1 has at least eight ligand-binding domains and motifs (*Duman et al., 2016*). Unfortunately, there have been fewer ligands reported for BAI1 than ligand-binding domains, but those include phosphatidylserine (*Park et al., 2007*), bacterial lipopolysaccharide (*Das et al., 2011*), and $\alpha_v\beta_5$ integrin (*Koh et al., 2004*). Additional BAI1 ligands relevant for neuronal development and function remain to be identified. Moreover, other changes in BAI1 might affect its signaling, including its level, which does appear to increase shortly before and during the process of growth arrest (*Figure 7a*). Clearly, a full description of BAI1's levels, ligands, and signaling outcomes would be helpful for understanding these phenomena.

## Conclusion

We identified a new mechanism for restricting dendrite growth that connects the A-GPCR BAI1 to the small GTPase RhoA through Bcr. We provide evidence that directly correlates RhoA activation state to dendrite behavior, and show that BAI1 is required to maintain this relationship. In building on the rapidly growing portraits of A-GPCR function, our results suggest that BAI1 integrates signals and can 'select' pathways to mediate differing outcomes. These results might be generalizable to other A-GPCRs, perhaps especially the closely related BAI2/ADGRB2 and BAI3/ADGRB3 (*Shiratsuchi et al., 1997*). However, even by itself, this pathway has great potential significance. Could activation of this dendritic restriction pathway correct dendritic hypertrophy in ASD models and patients? Could inhibition thereof correct hypotrophy in other ASD patients? Is dendrite retraction in neurodegenerative disease caused by hyperactivation of this or other dendritic restriction mechanisms? The answers to these questions will have great significance in addressing the challenges of human mental health.

# Materials and methods

**Key resources table**

| Reagent type (species) or resource | Designation | Source or reference | Identifiers | Additional information |
|---|---|---|---|---|
| Gene (*Homo sapiens*) | ADGRB1 | NA | Gene ID:575 | |
| Gene (*Rattus norvegicus*) | ADGRB1 | NA | Gene ID:362931 | |
| Gene (*H. sapiens*) | BCR | NA | Gene ID:613 | |
| Gene (*Mus musculus*) | BCR | NA | Gene ID:110279 | |
| Gene (*M. musculus*) | RHOA | NA | Gene ID:11848 | |
| Strain (*M. musculus*) | RhoAflox/flox | PMID:23825607 | | |
| Strain (*M. musculus*) | Bcr KO | Jackson Labs | Stock no. 026396 | |
| Strain (*R. norvegicus*) | Long-Evans | Envigo, Charles River | HsdBlu:LE (E); Strain 006 (CR); RRID:RGD_2308852 | Timed pregnant females |
| Cell line (*Cercopithecus aethiops*) | Cos-7 | ATCC | CRL-1651; RRID:CVCL_0224 | |
| Antibody | anti-BAI1 (C-terminal) (rabbit polyclonal) | PMID:23595754 | 2525 | 0.001 mg/ml for WB, IHC, ICC; 1 µg/1.3E6 neurons for IP |
| Antibody | anti-BAI1 (N-terminal) (rabbit polyclonal) (H-270) | Santa Cruz Biotechnology | Santa Cruz :sc-66815; RRID:AB_2062912 | (1:500) |

*Continued on next page*

Continued

| Reagent type (species) or resource | Designation | Source or reference | Identifiers | Additional information |
|---|---|---|---|---|
| Antibody | anti-Bcr (rabbit polyclonal) (N-20) | Santa Cruz Biotechnology | Santa Cruz:sc-885; RRID:AB_2274682 | (1:500) |
| Antibody | anti-Abr (mouse monoclonal) (*Lin et al., 2013*) | Santa Cruz Biotechnology | Santa Cruz :sc-135821; RRID:AB_2221350 | (1:1000) |
| Antibody | anti-c-myc (mouse monoclonal) (9E10) | Santa Cruz Biotechnology | Santa Cruz:sc-40; RRID:AB_627268 | (1:1000) |
| Antibody | anti-BAI2 (extracellular) (rabbit polyclonal) | Alomone Labs | Alomone:ABR-022; RRID:AB_2756544 | (1:1000) |
| Antibody | anti-BAI3 (rabbit polyclonal) | Sigma-Aldrich | Sigma-Aldrich: HPA015963; RRID:AB_1845263 | (1:500) |
| Antibody | anti-actin (clone C4) (mouse monoclonal) | Millipore | Millipore:MAB1501; RRID:AB_2223041 | (1:10,000) |
| Antibody | Cy3-anti-rabbit secondary | Jackson ImmunoResearch Laboratories | Jackson ImmunoResearch: 115165003; RRID: AB_2338000 | (1:500) |
| Antibody | Cy3-anti-mouse secondary | Jackson ImmunoResearch Laboratories | Jackson Immuno Research:115165146; RRID:AB_2338690 | (1:500) |
| Antibody | HRP-anti-rabbit secondary | Millipore | Millipore:401393; RRID:AB_437797 | (1:20,000) |
| Antibody | HRP-anti-mouse secondary | Millipore | Millipore:401215; RRID:AB_10682749 | (1:20,000) |
| Recombinant DNA reagent | pCMV-EGFP | Connie Cepko | RRID:Addgene_11153 | |
| Recombinant DNA reagent | pcDNA3.1-mRuby2 | Michael Lin | RRID:Addgene_40260 | |
| Recombinant DNA reagent | pCx-EGFP | PMID:23595754 | | |
| Recombinant DNA reagent | pRaichu-Raichu-Rac1 | PMID:16472667 | 1034X | |
| Recombinant DNA reagent | pRaichu-Raichu-RhoA | PMID:16472667 | 1298X | |
| Recombinant DNA reagent | pRaichu-Raichu-Rac1-DN | PMID:16472667 | 1013X | |
| Recombinant DNA reagent | pRaichu-Raichu-Rac1-CA | PMID:16472667 | 1012X | |
| Recombinant DNA reagent | pSuper | Oligoengine | Oligoengine: VEC-pBS-0002 | PMID:11910072 |
| Recombinant DNA reagent | pSuper-shRNA1 | PMID:23595754 | | 5'-GCCCAAATACAGCATCAACA-3' |
| Recombinant DNA reagent | pSuper-shRNA2 | PMID:23595754 | | 5'-CCCGGACCCTCGTCGTTAC-3' |
| Recombinant DNA reagent | pcDNA3.1-BAI1 | PMID:9533023 | | |
| Recombinant DNA reagent | pcDNA3.1-BAI1•ΔTEV | PMID:23595754 | | |
| Recombinant DNA reagent | pcDNA3.1-BAI1•ΔN-term. | PMID:30120207 | | |

*Continued*

| Reagent type (species) or resource | Designation | Source or reference | Identifiers | Additional information |
|---|---|---|---|---|
| Recombinant DNA reagent | pcDNA3.1-BAI1•RKR | PMID:17960134 | | K. Ravichandran (University of Virginia) |
| Recombinant DNA reagent | pcDNA3.1-BAI1•ΔGPCR | this paper | | available upon request from K. Tolias |
| Recombinant DNA reagent | pcDNA3.1-BAI1•ΔPRR | this paper | | available upon request from K. Tolias |
| Recombinant DNA reagent | pcDNA3.1 (+) | ThermoFisher Scientific | ThermoFisher Scientific:V790-20 | |
| Recombinant DNA reagent | pCMVmyc-Bcr | PMID:24960694 | | |
| Recombinant DNA reagent | pCMVmyc-Abr | PMID:24960694 | | |
| Recombinant DNA reagent | pCMVmyc-Bcr-GAP-dead | PMID:24960694 | myc-Bcr-GAP-dead | (R1090A) |
| Recombinant DNA reagent | pCMVmyc-Bcr-GEF-dead | this paper | Myc-Bcr-GEF-dead | (N689A/E690A) available upon request |
| Recombinant DNA reagent | pCMV-myc | Clontech | Clontech:635689 | |
| Recombinant DNA reagent | pGEX-4T1 | Amersham | Amersham:27458001 | |
| Recombinant DNA reagent | pGEX-4T1-BAI1-C-term. | this paper | | available upon request from K. Tolias |
| Sequence-based reagent | 5'-GGAGGGCAGAGGCTGTGAG-3' | this paper | BAI1 forward primer | available upon request from K. Tolias |
| Sequence-based reagent | 5'-GCAGAGGCTCCAGGGTGAC-3' | this paper | BAI1 reverse primer | available upon request from K. Tolias |
| Sequence-based reagent | 5'-ATGACCGACTTCGAGAAGGACG-3' | PMID:15225653 | BAI2 forward primer | |
| Sequence-based reagent | 5'-CTGCACGTCATCAGCGGAAG-3' | this paper | BAI2 reverse primer | available upon request from K. Tolias |
| Sequence-based reagent | 5'-TAACCGGCCAGCAGTGTGAAG-3' | PMID:24567399 | BAI3 forward primer | |
| Sequence-based reagent | 5'-CATTCCATCACCTGCCAGCAT C-3' | this paper | BAI3 reverse primer | available upon request from K. Tolias |
| Sequence-based reagent | 5'-GATGATATCGCCGCGCTCGTC-3' | PMID:15225653 | actin forward primer | |
| Sequence-based reagent | 5'-AGCCAGGTCCAGACGCAGGAT-3' | PMID:15225653 | actin reverse primer | |
| Commercial assay or kit | UltraLink BioSupport | ThermoFisher Scientific | ThermoFisher Scientific:53110 | |
| Chemical compound, drug | tetradotoxin | Tocris | Tocris:1078 | |
| Chemical compound, drug | narciclasine | Tocris | Tocris:3715/1 | |
| Chemical compound, drug | D-AP5 | Tocris | Tocris:0106/1 | |
| Chemical compound, drug | Y-27632 dihydrochloride | Tocris | Tocris:TB1254-GMP/10 | |
| Chemical compound, drug | Potassium chloride | ThermoFisher Scientific | ThermoFisher Scientific:p217 | |
| Software, algorithm | Imaris | Oxford Instruments | RRID:SCR_007370; RRID:SCR_007366 | Version 9.2.1 |

*Continued on next page*

*Continued*

| Reagent type (species) or resource | Designation | Source or reference | Identifiers | Additional information |
|---|---|---|---|---|
| Software, algorithm | Fiji | PMID:22743772 | RRID:SCR_002285 | |
| Software, algorithm | Sholl analysis plugin | PMID:25264773 | | |
| Software, algorithm | PIXFRET | PMID:16208719 | | |
| Software, algorithm | riFRET | PMID:19591240 | | |
| Software, algorithm | Zen | Carl Zeiss. Jena, Germany | RRID:SCR_013672 | Version 2.3 |
| Software, algorithm | Prism | Graph Pad | RRID:SCR_002798 | Version 8 |

## Animals

Neuronal cultures for rats were prepared from Long-Evans rats (Evigo and Charles River, Huntingdon, UK and Wilmington, MA) at E18. For in vivo experiments, in utero electroporation was performed on E14 ICR mice (Charles River). RhoA$^{flox/flox}$ and Bcr$^{KO}$ mice have already been described (*Mulherkar et al., 2013*; *Mulherkar et al., 2014*; *Um et al., 2014*). All animals were maintained in Baylor College of Medicine's animal facilities, which are fully accredited by the Association for Assessment and Accreditation of Laboratory Animal Care International. All procedures were approved by the Animal Care and Use Committee of BCM (IACUC) in accordance with applicable legislation.

## DNA Constructs

pCMV-EGFP was used in mouse and rat cultures for morphological analysis, as we described previously (*Duman et al., 2013*). In some experiments we used pcDNA3.1-mRuby2 for morphological analysis, and pCx-EGFP was used to drive expression of EGFP in vivo after in utero electroporation: both were obtained from M. Rasband (Baylor College of Medicine). We used the following FRET-based Rho-GTPase activity reporters: pRaichu-RaichuEV-Rac1 (2248X) (*Komatsu et al., 2011*), pRaichu-RaichuRhoA (1298X) (*Nakamura et al., 2006*), and RhoA-FLARE (*Pertz et al., 2006*). The former two were obtained from M. Matsuda (University of Kyoto) and the last from K. Hahn (University of North Carolina). The two shRNAs against BAI1 used target non-overlapping sequences within the *Adgrb1* sequence subcloned into pSuper (Oligoengine, Seattle, WA) and were described previously (*Duman et al., 2013*). The sequences were 5'-GCCCAAATACAGCATCAACA-3' for shRNA1 (corresponding to bps 4296–4318 in rat *Adgrb1* mRNA) and 5'-CCCGGACCCTCGTCG TTAC-3' for shRNA2 (corresponding to bps 1116–1134 in rat *Adgrb1* mRNA). tdTomato-Cre was obtained from Edward Boyden. Human BAI1 was obtained from Y. Nakamura (University of Tokyo). We described BAI1•ΔTEV (*Duman et al., 2013*) and BAI1•ΔN-term. (*Tu et al., 2018*) previously. BAI1•RKR was obtained from K. Ravichandran (University of Virginia). BAI1•ΔGPCR and BAI1•ΔPRR were produced for this study using standard cloning techniques. The cDNA sequence between the first transmembrane domain of the BAI1-GPCR and the end of the seventh transmembrane domain was removed in the BAI1•ΔGPCR mutant. The cDNA encoding residues 1407–1430, a stretch containing 19 prolines in 24 amino acids, was removed in the BAI1•ΔPRR. pcDNA3.1 (+) (Invitrogen, Waltham, MA) was the host vector for all BAI1 constructs and was also used to balance DNA amounts in transfections. Myc-Bcr, myc-Abr, and myc-Bcr-GAP-dead (R1090A) were as described previously (*Um et al., 2014*). Myc-Bcr-GEF-dead (N689A/E690A) was created by subcloning the mutated Bcr sequence from ECFP-Bcr NE/AA, obtained from Nora Heisterkamp (Addgene, plasmid #36418) into pCMV-myc. All constructs were sequence verified.

## Antibodies and drugs

An anti-BAI1 rabbit polyclonal was generated (Covance, Princeton, NJ) against amino acids 1180–1584 of human BAI1 and affinity purified using the antigen purified as a thrombin-cleaved GST-

fusion coupled to UltraLink BioSupport (Thermo Scientific, Waltham, MA). We described this antibody previously (*Duman et al., 2013*); BAI1 is only detected by this antibody in Western blots of Cos-7 cells when they are exogenously expressing BAI1. The immunoreactivity of this antibody for neurons is strongly decreased by BAI1 Kd. This antibody was used at 0.001 mg/ml for IHC, ICC, and Western blotting and was reused at least 5X. For immunoprecipitation, 1 µg per $1.3 \times 10^6$ neurons in culture was used. Rabbit anti-BAI11 N-terminal polyclonal (H-270, sc-66815), rabbit anti-Bcr polyclonal (N-20, sc-885), mouse anti-Abr monoclonal $IgG_1$ (24, sc-135821), and mouse monoclonal anti-c-myc $IgG_1$ (9E10, sc-40) were obtained from Santa Cruz Biotechnology (Dallas, TX) and used at 1:1000 dilution for Westerns and ICC. DMSO (D2650) was obtained from Sigma (St. Louis, MO). Tetradotoxin, narciclasine, D-AP5, and Y-27632 dihydrochloride were obtained from Tocris (Minneapolis, MN). Potassium chloride (P217) was obtained from Fisher Scientific (Pittsburgh, PA).

## In vivo assessment of BAI1 function

EGFP and empty or shRNA-containing pSuper was introduced into E14 mice via in utero electroporation as described (*Hedstrom et al., 2007*). Pups were euthanized at P21. Brains were dissected and fixed with 4% paraformaldehyde (Sigma) at 4 ˚C for 1 hr on ice and equilibrated in ice-cold 20% w/v sucrose (Sigma). 30 µm sections were cut and used for immunostaining in PBS (Corning, Corning, NY) containing 0.3% Triton X-100 (USB, Salem, MA) and 5% BSA (Fisher Scientific). Primary antibodies against BAI1 were applied for 2–3 days and Cy3-labelled goat anti-rabbit IgG (Jackson Immunoresearch, West Grove, PA) was applied for 1 day. Sections were mounted in VectaShield (Vector Laboratories, Burlingame, CA) 100 µm coronal sections were cut from regions containing hippocampus for assessing dendrite arborization. After collecting image stacks with a Δz of 0.5 µm, dendritic arbors of EGFP-expressing pyramidal neurons from dorsolateral CA1 were reconstructed as below. Only neurons whose cell bodies were centered within 10 µm of the center of the stack z-dimension were analyzed. Reconstruction was limited to single sections because the relatively high local levels of transfection made it impossible to positively assign unassociated dendrites, even in serial sections.

## Neuronal culture, transfection, and immunostaining

Rat hippocampal cultures (used in *Figures 1i–k*, *2–6*, *7d–i* and *8d–f* and *Figure 1—figure supplement 1*; *Figure 1—figure supplement 2*; *Figure 2—figure supplement 1*; *Figure 3—figure supplement 2*; *Figure 4—figure supplement 1*; *Figure 4—figure supplement 2*; *Figure 5—figure supplement 1*; *Figure 7—figure supplement 1*) were prepared from E18 rats. Hippocampi were dissected out, dissociated with papain, washed with trypsin inhibitor (Sigma) and seeded onto nitric acid-washed #1.5 glass (*Figures 1*, *3a–c*, *4* and *5* and *Figure 1—figure supplement 1*; *Figure 2—figure supplement 1*; *Figure 3—figure supplement 2—source data 1*; *Figure 4—figure supplement 1*; *Figure 4—figure supplement 2*; *Figure 5—figure supplement 1*) or cell culture-treated plastic (*Figures 2*, *3d–i*, *6*, *7a–c,d–i* and *8d–f* and *Figure 1—figure supplement 2*; *Figure 3—figure supplement 2*; *Figure 4—figure supplement 1*; *Figure 5—figure supplement 1*; *Figure 7—figure supplement 1*) coated with 20 µg/ml poly-D-lysine (Corning) and 3 µg/ml laminin (Corning) at $3.0 \times 10^5$ neurons/ml in Neurobasal medium (Invitrogen) supplemented with B27 (Invitrogen), 2 mM glutamine (Thermo Scientific), and 100 U/ml penicillin/streptomycin (Thermo Scientific). Culture medium was changed on DIV 1. Mouse hippocampal neurons (*Figure 8a–c* and *Figure 3—figure supplement 3*) were prepared from P0 or P1 pups, genotyped by PCR, plated onto pre-coated one glass coverslips (Corning BioCoat), and cultured in the same medium as above only with Neurobasal A medium (Invitrogen). Neurons were transfected on 7 DIV in early experiments (*Figures 1*, *3a–c* and *5a–b,f–h* and *Figure 1—figure supplement 1*; *Figure 2—figure supplement 1*; *Figure 3—figure supplement 2*; *Figure 5—figure supplement 1*), but on 6 DIV for all other experiments (*Figures 2*, *3d–i*, *4*, *5c–e,f–h* and *6–8* and *Figure 3—figure supplement 2*; *Figure 3—figure supplement 3*; *Figure 4—figure supplement 1*; *Figure 4—figure supplement 2*; *Figure 5—figure supplement 1*; *Figure 7—figure supplement 1*). This difference appeared to matter at early stages, as 7 DIV-transfected BAI1 knockdown (Kd) neurons were the same length as controls on 10 DIV, but 6 DIV-transfected BAI1 Kd neurons were shorter than control neurons at 10 DIV. By 21 DIV, it did not matter whether the neurons had been transfected on 6 or 7 DIV. We utilized a modified calcium phosphate transfection protocol using 1.0 µg total DNA/24-well plate well equivalent as

previously (*Duman et al., 2013*). For BAI1 overexpression, 0.2 μg DNA/24-well plate well equivalent was used. Use of higher levels resulted in an apparent dominant negative effect. Mouse cultures were prepared from P0 pups in the same manner as for rats, plated on glass, and transfected on 6 DIV as above. At the indicated times or at the end of longitudinal time courses, cultured neurons were fixed with 4% paraformaldehyde and 20% sucrose and stained in PBS containing 5% BSA, 15% goat serum (Invitrogen), and 0.1% Triton X-100. Cy3-conjugated goat anti-rabbit IgG (Jackson) was used as the secondary antibody.

## Tissue culture and transfection

Cos-7 cells were cultured in DMEM (Corning) supplemented with penicillin/streptomycin and 10% fetal bovine serum on cell culture-treated plastic. Cells were passaged with trypsin 1-2X per week. Transfection was by the calcium phosphate method as above for neurons or by LipofectAmine 2000 (ThermoFisher) as per manufacturer's instructions. A MycoAlert mycoplasma test conducted by the Cell Biology Tissue Culture Core at Baylor College of Medicine confirmed the absence of mycoplasmal infection.

## Surface staining of BAI1

Cos-7 cells were transfected with pCMV- EGFP and wild-type human BAI1 or the ΔTEV, RKR, ΔGPCR, or ΔPRR mutants contained in pcDNA3.1(+) using the calcium phosphate technique used for neurons above, the only difference being that the precipitates were not washed off of the cells. One day later, the cells were put on ice and live-stained with 1:500 anti-BAI1 N-terminal polyclonal or anti-BAI1 C-terminal polyclonal in PBS containing 5% BSA for 2 hr. The cells were then washed 4X with ice-cold PBS and fixed with 4% paraformaldehyde and 20% sucrose and washed 3X with PBS. Cells were stained overnight with 1:500 Cy3-conjugated goat anti-rabbit IgG as above. After washing 3X, cells were stained overnight with anti-BAI1 C-terminal antibody and Alexa 568-conjugated goat anti-rabbit IgG as in the neurons above.

## RNA isolation and real-time RT-PCR

Total RNA was isolated from DIV 16 rat primary cultures using the Direct-zol TM RNA MiniPrep Plus kit (Zymo R2070) and reverse transcribed using the Quantiscript Reverse Transcriptase and random primers (Qiagen, 205311) following the manufacturers' protocols. All rtPCR reactions were carried out using GoTaq Green PCR Master Mix (Promega, M7128) and a BioRad C1000 Touch thermocycler with the following conditions: 5 min at 94°C, followed by 40 cycles of 30 s at 94°C, 30 s at 60°C, and 45 s at 72°C, then a final 5 min at 72°C. The amplification products were visualized on 1% agarose gels under UV epifluorescence following ethidium bromide staining. Primers used include 5′-ATGACCGACTTCGAGAAGGACG-3′ (forward) and 5′-TCTGCGGCATCTGGTCAATGTG-3′ (reverse) for BAI1; 5′-ATGACCGACTTCGAGAAGGACG-3′ (forward) and 5′-CTGCACGTCATCAGCGGAAG-3′ (reverse) for BAI2; 5′-TAACCGGCCAGCAGTGTGAAG-3′ (forward) and 5′-CATTCCATCACCTGCCAGCAT C-3′ (reverse) for BAI3; and 5′-GATGATATCGCCGCGCTCGTC-3′ (forward) and 5′-AGCCAGGTCCAGACGCAGGAT-3′ (reverse) for actin. Reverse primers for BAI2 and BAI3 were designed using Snapgene.

## SDS-PAGE and western blotting

Cells were lysed in NP40 lysis buffer (50 mM Tris, pH 7.5, 150 mM NaCl, 1% NP-40, 1 mM EDTA, pH 8.0, 5% glycerol) with 1 mM DTT, Complete protease inhibitor mixture (Roche, 4693116001), and XpertPhosphatase Inhibitor Cocktail (GenDEPOT, P3200-001) 10 days after transfection. Proteins were then subjected to SDS-PAGE using 4–15% gradient gels (BioRad, 456–1084), followed by transfer to Immobilon-P (Millipore) PVDF membranes. All western blots were visualized on an ImageQuant LAS4000 (GE Lifesciences).

## Microscopy

Three microscopes were used to collect data: a Leica TCS SP2 confocal microscope (*Figures 1*, *3a–c* and *5a–b,f–h* and *Figure 2—figure supplement 1*; *Figure 3—figure supplement 2*; *Figure 5—figure supplement 1*), a Zeiss AxioObserver.1 attached to an Apotome (*Figures 2*, *3d–i*, *4*, *5c–e,f–h*, *6*, *7j–k* and *8a–c* and *Figure 3—figure supplement 2*; *Figure 3—figure supplement 3*;

*Figure 4—figure supplement 1*; *Figure 5—figure supplement 1*; *Figure 7—figure supplement 1*), and a Zeiss LSM 880 operating in confocal mode (*Figures 7d–f* and *8d–f* and *Figure 3—figure supplement 1*; *Figure 4—figure supplement 1*; *Figure 4—figure supplement 2*; *Figure 5—figure supplement 2*). Z-stacks ($\Delta z$ = 1 µm) from sections cut from intact brains (*Figure 1*) were imaged with a 63 × oil immersion lens and the pinhole set at one airy. Whole-arbor images of cultured neurons were collected with 10 × objectives, while high magnification images (*Figure 5a–e* insets and Supplementary Data 3f-h, 8) employed 63 × objectives and were collected as Z-stacks ($\Delta z$ = 0.2 µm). Fixed cultures were mounted in FluorSave (EMD Millipore, Burlington, MA). For longitudinal imaging, neurons or Cos-7 cells were switched into 0.2 µm-filtered HEPES-containing 1.5 × ACSF (124 mM NaCl, 3.0 mM KCl, 1.3 mM $NaH_2PO_4 \cdot H_2O$, 3.0 $MgCl_2$, 10.0 mM dextrose, 10 mM HEPES (pH 7.4), and 3.0 mM $CaCl_2$) (all reagents from Sigma-Aldrich), and a memory stage on the AxioObserver was employed to revisit the same neurons. FRET data collected with Leica employed the following settings: excitation 458 nm, emission 530–600 nm (FRET); excitation 458 nm, emission 470–500 nm (CFP); and excitation 514 nm, emission 530–600 nm (YFP). FRET data was collected with the AxioObserver using the following filter sets: excitation BP 436/20, beamsplitter FT455, emission BP 535/30 (FRET); excitation BP 436/25, beamsplitter FT455, emission BP480/40 (CFP); and excitation BP 500/25, beamsplitter FT515, emission BP 535/30 (YFP). Standard EGFP and Cy3 confocal settings and filter sets were employed when appropriate on all instruments.

## Data extraction and analysis

Experimental conditions of samples were blinded to data collectors and analyzers for most experiments. In some longitudinal studies, samples were not blinded but neurons were selected very early in development before any differences between the conditions were apparent (5 DIV). Removal of neurons from these analyses was not allowed with the single exception of neurons that died prior to the end of the time course. For fixed neurons, neurons were excluded if staining indicated very high overexpression of BAI1 or failure of BAI1 Kd. Both exclusions were rare. For dendritic arbors from in vivo hippocampus, z-stacks were used to create 3D reconstructions in Imaris 7.1 (Bitplane, Belfast, UK), and for the spine measurements in *Figure 3—figure supplement 2*, Imaris 9.1 was used. All data reported in *Figure 1* were extracted from these reconstructions. For single-plane reconstructions, we used the NeuronJ plugin of ImageJ (NIH); again, all morphological data were extracted from these models. Sholl analyses were performed using the ImageJ Sholl plugin. BAI1 expression levels in *Figure 1—figure supplement 2H* were extracted from z-stacks using the Spots object function in Imaris, corrected for background. FRET data were analyzed using the PIX-FRET (*Feige et al., 2005*) or RiFRET (*Roszik et al., 2009*) plugins for ImageJ. Absolute FRET images were divided by images of the directly excited acceptor to correct FRET images for differences in probe expression. This was not necessary for images processes with RiFRET. These processed values, restricted to certain regions of interest (ROI) at given times are referred to as $F_{ROI,t}$ in the formulae below. Dendrite tips were defined as the last 1 µm of dendritic shaft. RhoA activation levels in the dendritic tips closely paralleled overall dendritic levels but were easier to extract. Dendrite branch points were defined as the region of the parent dendrite (defined as the dendrite with larger gauge after the branch) normal to the daughter dendrite and equal in depth to the base of the daughter dendrite. This method encompasses the specialized structures, such as Golgi outposts, that reside in dendritic branch points (*Quassollo et al., 2015*). Normalized FRET intensity in various regions of interest was collected as noted. Data were plotted using IgorPro (WaveMetrics, Lake Oswego, OR). The lookup table used to show relative differences in normalized FRET is shown in each relevant figure. To date, no calibration of the FRET probes used in this study has been published. Thus, the differences in FRET indicate a continuum of relative differences, i.e. higher and lower, but the exact numerical relationship between them is unknown. As a result, it was necessary to normalize to points fixed in time and space. For *Figures 3a-c*, *6*, *7j-k* and *Figure 3—figure supplement 2*; *Figure 7—figure supplement 1*, the reported FRET values were calculated as:

$$F_{norm} = F_{ROI,t} \div \bar{F}_{Cont.,Som.,t=i}$$

where $F_{norm}$ is the final FRET value, $F_{ROI,t}$ is the processed FRET value in a particular region of interest and $\bar{F}_{Cont.,Som.,t=i}$ is the average processed FRET values of all of the control somata at the initial time

point. For the data shown in *Figure 3e-i*, *Figure 3—figure supplement 1*, *Figures 5e-g*, *7a-c* the reported FRET values were calculated as:

$$F_{norm} = \left( F_{ROI,t} - \bar{F}_{Cont., Stab., t=i} \right) \div F_{Cont., Stab., t=i}$$

where $F_{norm}$ and $F_{ROI,t}$ are defined as above, and $\bar{F}_{Cont., Stab., t=i}$ is the average processed FRET value of the stable dendrites in control neurons at the initial time point or the whole cell at the start of the experiment for Cos-7 cell experiments. These data are uncalibrated, as are all measurements utilizing Raichu probes or RhoA-FLARE. % differences between measurements reflect normalized % changes, but these may not map linearly to changes in RhoA or Rac1 activation in cells.

## Collage assembly

For each collage, all images were taken from the same repeat of the experiment. In *Figure 8a*, the displayed images in each row come from the same animal. For the images of somatodendritic domains (*Figures 1i, 2a, f, 3a, e–f, 4a, f, 5a–e, 6a, 7d, 8a and d* and *Figure 1—figure supplement 1*; *Figure 2—figure supplement 1*; *Figure 3—figure supplement 2*; *Figure 3—figure supplement 3*; *Figure 4—figure supplement 2*; *Figure 5—figure supplement 1*) all axons and neurons aside from those chosen were cropped out of the image and the image was thresholded for ease of viewing. In all cases, the same display parameters (background correction, brightness, contrast, gamma) were used for all panels within a collage.

## Statistical methods

All data are shown ± s.e.m., except for the insets in *Figure 3i* and *Figure 7i* in which 95% confidence intervals are shown. Most data sets were initially subjected to ANOVA with the appropriate dimensionality (1-, 2-, or 3-way). If F exceeded $F_{crit}$ with a P value of less than 0.05 for any of the tests returned by the ANOVA, we rejected the null hypothesis and proceeded to *post hoc* analysis. Tukey's *post hoc* analysis was used for pairwise comparisons when appropriate. Normality was assumed for all data sets, and data spot checks confirmed this. In a few cases with only two conditions Student's T-test (2-tailed) was used. Differences between conditions were considered significant if Tukey's or Student's P values were less than 0.05. Data pairs within a single graph may be assumed to be statistically the same unless otherwise noted. In figure legends, *N* refers to the number of biologically independent preparations, i.e. the number of individual timed pregnant rats from which cultures were derived, individual mice, or completely independent cultured cell repeats of an experiment. The numbers of individual cells, dendrites, etc. for each condition (i.e., technical repeats) are reported in the figure legends. The fits in *Figure 3i* were obtained via linear regression of automatically binned data. Differences between the slopes of these fits were considered significant if both (i) separation between the values of the slopes exceeded the errors associated with the fitted slopes and (ii) the binned data points from which the slopes were derived showed significant differences across the Δlength continuum. No explicit power analysis was used to determine sample size, but all experiments have at least three independent biological repeats ($N \geq 3$); many experiments have more than three repeats due to low numbers of individual neurons within one or more conditions within one or more repeats. All statistical analyses were performed in IgorPro, except for 3-factor ANOVAs, which were performed in Prism GraphPad (San Diego, CA).

## Acknowledgements

We thank M Matsuda (University of Kyoto), K Hahn (University of North Carolina), Y Nakamura (University of Tokyo), K Ravichandran (University of Virginia), and M Rasband (Baylor College of Medicine) for providing DNA constructs. We also thank N Heisterkamp (Children's Hospital Los Angeles) for creating and sharing the Bcr KO mice. We also thank M Taylor, K Firozi, T Munjal, K Um, and B Schwechter for technical assistance. This work was supported by National Institute of Neurological Disorders and Stroke grant R01NS062829 (KFT), National Institute of Mental Health grants R01MH109511 (KFT) and K01MH089112 (JGD), and the Mission Connect-TIRR Foundation (KFT).

## Additional information

### Funding

| Funder | Grant reference number | Author |
| --- | --- | --- |
| National Institute of Neurological Disorders and Stroke | R01NS062829 | Kimberley F Tolias |
| National Institute of Mental Health | R01MH109511 | Kimberley F Tolias |
| National Institute of Mental Health | K01MH089112 | Joseph G Duman |
| Mission Connect-TIRR Foundation | | Kimberley F Tolias |

The funders had no role in study design, data collection and interpretation, or the decision to submit the work for publication.

### Author contributions

Joseph G Duman, Conceptualization, Data curation, Formal analysis, Supervision, Funding acquisition, Validation, Investigation, Visualization, Methodology, Writing—original draft, Writing—review and editing; Shalaka Mulherkar, Investigation; Yen-Kuei Tu, Christopher P Tzeng, Resources, Validation, Investigation; Kelly C Erikson, Vasilis C Mavratsas, Validation, Investigation; Tammy Szu-Yu Ho, Resources; Kimberley F Tolias, Conceptualization, Data curation, Supervision, Funding acquisition, Methodology, Project administration, Writing—review and editing

### Author ORCIDs

Joseph G Duman (iD) https://orcid.org/0000-0002-6711-0656
Shalaka Mulherkar (iD) https://orcid.org/0000-0001-8736-527X
Kimberley F Tolias (iD) https://orcid.org/0000-0002-2092-920X

### Ethics

Animal experimentation: This study was performed in strict accordance with the recommendations in the Guide for the Care and Use of Laboratory Animals of the National Institutes of Health. All of the animals were handled according to approved institutional animal care and use committee (IACUC) protocols (#AN-4365) of Baylor College of Medicine.

### Decision letter and Author response

Decision letter https://doi.org/10.7554/eLife.47566.060
Author response https://doi.org/10.7554/eLife.47566.061

## Additional files

### Supplementary files

• Transparent reporting form
DOI: https://doi.org/10.7554/eLife.47566.058

### Data availability

All data generated or analysed during this study are included in the manuscript and supporting files. Source data files have been provided for all relevant figures.

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
