## [Decision Letter]

Thank you for submitting your article "The adhesion-GPCR BAI1 shapes dendrites via contact-dependent, Bcr-mediated RhoA activation causing late growth arrest" for consideration by *eLife*. Your article has been reviewed by three peer reviewers, and the evaluation has been overseen by a Reviewing Editor and Marianne Bronner as the Senior Editor. The reviewers have opted to remain anonymous.

The reviewers have discussed the reviews with one another and the Reviewing Editor has drafted this decision to help you prepare a revised submission.

Summary:

This manuscript reports the mechanisms and functions of the adhesion GPCR BAI1 in dendritic arborization. Loss and gain of function studies indicate that BAI1 negatively regulates dendritic growth. The loss-of-function experiments are well controlled with LOF done with two independent shRNAs causing similar effects, rescue by re-expression of an shRNA resistant BAI1, and parallel in culture and in vivo studies. Interestingly, the time course analysis conducted following knockdown shows that BAI1 is required at a late stage (>14DIV) when dendrites are expanding rather than when they first emerge. The authors then address the signaling mechanisms downstream of BAI1 that mediate restriction of dendritic arborization and provide comprehensive data implicating Rho GTPases and a coupling of BAI1 to Bcr.

The reviewers felt that the manuscript was detailed and rigorous, and that it will have a high impact in the field of RhoA dependent processes that mediate control of neuronal development. We had concerns about a few points that we felt could be fully addressed upon revision with the addition of a few pieces of new data together with textual revisions.

Essential revisions:

1) Most FRET probes are used to measure signaling dynamics on the second and minute time scales. This study uses these FRET probes to quantify RhoGTPase activity at a single timepoint each day. It's unclear if these probes provide an effective readout over this timescale, and whether they can used to quantify absolute RhoGTPase activity. The authors would strengthen their interpretation of these data if they could address the following points: Is there a way to selectively and reproducibly activate BAI1 to show a rise in Raichu-RhoA or RhoA-Flare activity over seconds or minutes timeframe? Is there an alternative method to measure Rac and Rho activity to confirm FRET probe results?

2) Some controls need to be added to the shRNA experiments. In the Discussion, the authors cite previous work showing no dendritic arbor defects in BAI1 knockout mice. They suggest this lack of phenotype may be either due to compensation by BAI2/BAI3 or relate to competitive advantages that are seen in sparse knockdown but not in full knockouts. Have the authors confirmed that their shRNAs do not affect the levels of BAI2/BAI3? Considering the differences between the knockout and the shRNA experiments, this seems like a critical piece of data to add. Reviewers also asked for quantification of BAI1 knockdown at the protein level by western either from neurons or heterologous cells overexpressing BAI1. The BAI1 shRNA reagents are essential to the study and likely will be important to others who may later request it. Finally one reviewer noted that they could not find any description of the shRNA. What is the sequence? Where is it targeting BAI1? Please clarify.

3) The authors test the hypothesis that RhoA mediates BAI1's effect on dendrite growth in vitro using hippocampal neurons from RhoA conditional mice treated with tdTomato-Cre (Figure 3—figure supplement 3). Did the authors attempt to perform this experiment in vivo? It would be highly significant, if possible, to show whether electroporating tdTomato-Cre into RhoA^flox/flox^ mice at E18 would lead to a similar BAI1-mediated dendritic overgrowth phenotype at P21 as shown in Figure 1.

4) Do the mutations the authors make in BAI1 affect the surface localization of the protein? It looks from the Materials and methods as though the BAI1 antibody targets an intracellular domain and the GFP tag is also on the intracellular side. It should be clarified if data are already shown that address this point, or it needs to be experimentally addressed.

5) Finally, all three reviewers raised concerns about the interpretation of the dendritic avoidance observation. These concerns included whether the 2D growth of neurons used here is relevant to the mechanisms of dendritic avoidance that are usually studied in vivo, how the data fit into the rest of the literature on this important (and well-studied topic), and the fact that these data do not really answer the question of what normally activates BAI1. Because we felt the experiments needed to strengthen this section were beyond the scope of the current study we suggest instead that the authors consider removing this section.

---

## [Author Response]

Essential revisions:1) Most FRET probes are used to measure signaling dynamics on the second and minute time scales. This study uses these FRET probes to quantify RhoGTPase activity at a single timepoint each day. It's unclear if these probes provide an effective readout over this timescale, and whether they can used to quantify absolute RhoGTPase activity. The authors would strengthen their interpretation of these data if they could address the following points: Is there a way to selectively and reproducibly activate BAI1 to show a rise in Raichu-RhoA or RhoA-Flare activity over seconds or minutes timeframe? Is there an alternative method to measure Rac and Rho activity to confirm FRET probe results?

The reviewers are correct that FRET probes are most often used for short-term (seconds to 10s of minutes) measurements. We ourselves have used an A-GPCR-specific activation mechanism (the Stachel peptide) to show BAI1-mediated Rac1 activation over such a time course (Tu et al., 2018). However, we do not yet know the ligand(s) that promote RhoA activation via BAI1. One of the advantages of the construction of single-molecule probes like the Raichu and FLARE probes used in our manuscript is that, in principle, they are able to be calibrated and to provide reproducible, quantitative readouts over long periods of time. This means that if the imaging conditions (e.g., optics, lasers, detectors, temperature, media) do not change, the readout from the probe should be reproducible for any condition measured by and within the range of the probe. (That the Rho-GTPase activation probes have not been calibrated has more to do with creating accurate calibration conditions than with the probes themselves. Calcium probes created on the same backbone have been successfully calibrated thanks to the availability of calcium buffers, e.g. Palmer et al., 2004.) Moreover, Raichu-type probes have been used to record measurements for longer periods ranging from 10 hours to at least eight days (Kim et al., 2014; Konopa et al., 2016; Timpson et al., 2011).

Having said this, we did want to show that our measurements were stable over longer time frames than those more commonly used and have included these measurements in Figure 3—figure supplement 1. We show 24-hour stability for Raichu-RhoA probes mutated to give specific outcomes, for normal Raichu-RhoA in cells expressing RhoA GAPs and GEFs, and for normal Raichu-Rac1 in cells expressing Rac1 GAPs and GEFs. We also show that within the experiments shown in Figure 3D-I, short-term measurements are stable. Together, these data indicate that the FRET probes used reproducibly report expected results over at least 24 hours in Cos-7 cells. We were not able to perform these measurements in neurons because the dynamic nature of RhoA and Rac1 signaling during neuronal development and function makes it impossible to ‘clamp’ Rac1 and RhoA signaling states, even when regulatory proteins are overexpressed.

2) Some controls need to be added to the shRNA experiments. In the Discussion, the authors cite previous work showing no dendritic arbor defects in BAI1 knockout mice. They suggest this lack of phenotype may be either due to compensation by BAI2/BAI3 or relate to competitive advantages that are seen in sparse knockdown but not in full knockouts. Have the authors confirmed that their shRNAs do not affect the levels of BAI2/BAI3? Considering the differences between the knockout and the shRNA experiments, this seems like a critical piece of data to add. Reviewers also asked for quantification of BAI1 knockdown at the protein level by western either from neurons or heterologous cells overexpressing BAI1. The BAI1 shRNA reagents are essential to the study and likely will be important to others who may later request it. Finally one reviewer noted that they could not find any description of the shRNA. What is the sequence? Where is it targeting BAI1? Please clarify.

The reviewers are correct that these reagents are essential to this study, and we welcome the opportunity to address the points that they made. First, we have included the sequences of the shRNAs and indicated where they match to the rat BAI1 mRNA sequence in the Materials and methods section. Second, we have added new Figure 1—figure supplement 1 with additional original data to address shRNA function and possible compensation. These include: a cartoon depiction of the sites for both shRNAs within the rat BAI1 mRNA (panel a); RT-PCR showing BAI1, but neither BAI2 nor BAI3, knock down at the mRNA level in hippocampal neurons (panel b); Western blots showing BAI1, but neither BAI2 nor BAI3, knock down at the protein level in hippocampal neurons (panel c); quantification of the Western data in panel c (panel d); and serial dilution of control and BAI1 Kd samples showing linearity and loading controls in both pSuper- and shRNA-expressing neurons (panel e). We did these experiments using calcium phosphate transfection under conditions that affect 40-50% of hippocampal neurons. While other methods (e.g., LipofectAmine, virus) lead to potentially greater transfection levels, calcium phosphate better preserves the health of the neurons and is more similar to the conditions under which most of the experiments in this paper were performed. We also note that the primary focus of the BAI1 KO paper is spines and synapses, and there is no rigorous analysis of dendritic arbors in this paper (Zhu et al., 2015). It is therefore possible that we proposed a resolution of a problem that does not exist.

3) The authors test the hypothesis that RhoA mediates BAI1's effect on dendrite growth in vitro using hippocampal neurons from RhoA conditional mice treated with tdTomato-Cre (Figure 3—figure supplement 3). Did the authors attempt to perform this experiment in vivo? It would be highly significant, if possible, to show whether electroporating tdTomato-Cre into RhoA^flox/flox^ mice at E18 would lead to a similar BAI1-mediated dendritic overgrowth phenotype at P21 as shown in Figure 1.

We agree that this would be an interesting experiment that would extend the similarity of BAI1 and RhoA loss-of-function phenotypes from culture to an in vivo model. Under the best of circumstances, it would have been difficult to complete this experiment within the editors’ recommended time frame for revisions, but our colony of RhoA^flox/flox^ mice was low and staffing changes further complicated completion. In order to maintain a reasonable timeline, we have not performed this experiment. We have included additional references to the Discussion to paint a clearer picture of RhoA function in dendrite development. We hope that this will address at least some of the reviewers’ concerns.

We want to emphasize two points with regard to this experiment: (1) we are not opposed in principle to doing this experiment and could (and would) complete it if given more time. If the reviewers and editors decide that this experiment is mandatory for publication of this study in *eLife*, then we respectfully ask for a substantial extension in order to complete it. (2) We already present a large amount of data linking BAI1 to RhoA in dendrite growth arrest in this paper. In addition to the similarity between BAI1 Kd and RhoA KO phenotypes, we show that: (i) loss of BAI1 from hippocampal neurons leads to both dendrite overgrowth and lowered RhoA activation in two completely independent experiments using two different probes of RhoA activity; (ii) that BAI1 mutants that rescue dendrite overgrowth also rescue lowered RhoA activation, but that one mutant that fails to rescue dendrite overgrowth also fails to rescue lowered RhoA activation; (iii) BAI1 associates with the RhoA-GEF Bcr and this association strengthens at the time of dendritic growth arrest; (iv) Bcr is also required for normal dendritic growth arrest; (v) BAI1 and Bcr co-expression increases Bcr’s RhoA-GEF activity; (vi) Bcr’s RhoA-GEF activity is required for dendritic growth arrest; and (vii) use of pharmacological agents to activate RhoA or inhibit RhoA-dependent kinases reversed the effects of BAI1 gain- and loss-of-function, respectively.

4) Do the mutations the authors make in BAI1 affect the surface localization of the protein? It looks from the Materials and methods as though the BAI1 antibody targets an intracellular domain and the GFP tag is also on the intracellular side. It should be clarified if data are already shown that address this point, or it needs to be experimentally addressed.

The reviewers’ point is well taken. The mutants are not GFP-tagged, but we do have antibodies against both N-terminal (extracellular) and C-terminal (intracellular) epitopes of BAI1. This allowed us to demonstrate that wild-type BAI1, as well as the ∆TEV, RKR, ∆GPCR, and ∆PRR mutants localize to the surface of Cos-7 cells expressing them. The BAI1 ∆N-term. mutant does not contain the N-terminal epitope and so was not amenable to these experiments. However, in light of the inability of ∆N-term. to rescue the spine and synaptic phenotypes arising from BAI1 loss, we had already used a N-terminally flag-tagged version of this mutant to demonstrate its surface localization (Tu et al., 2018). In our revision, we have added these data as Figure 5—figure supplement 2, discussed the surface localization of BAI1-∆N-term. in the legend to these data, included our surface labeling protocol, and made textual changes to incorporate these data into the manuscript.

5) Finally, all three reviewers raised concerns about the interpretation of the dendritic avoidance observation. These concerns included whether the 2D growth of neurons used here is relevant to the mechanisms of dendritic avoidance that are usually studied in vivo, how the data fit into the rest of the literature on this important (and well-studied topic), and the fact that these data do not really answer the question of what normally activates BAI1. Because we felt the experiments needed to strengthen this section were beyond the scope of the current study we suggest instead that the authors consider removing this section.

We understand the reviewers’ concerns and have removed this section and all references to it in other parts of the paper, including the title, summary, Discussion, Materials and methods, figure legends, and supplementary data.